# Nonconvex Low-Rank Tensor Representation with Deep Priors for Multiview Subspace Clustering

Yao Fu [1]   Dong Hu [2]   Zhi Wang [1]

## Abstract

Multiview subspace clustering (MvSC) has shown remarkable potential in exploring underlying structures of high-dimensional data. However, existing MvSC methods still suffer from two shortcomings: (1) the commonly use of convex low-rank approximations inadequately capture high-order correlations across views, while sensitivity to noise and outliers degrades clustering performance, and (2) they lack the ability to preserve global correlations and local geometric patterns simultaneously. To address these issues, we propose a novel nonconvex regularized MvSC model with deep prior, which not only accurately characterizes the intrinsic low-rank structure and suppresses the effect of outliers, but also preserves local structural properties through deep networks. By mathematically analyzing the optimal solution of the optimization problem in our proposed model, we develop an efficient ADMM-based algorithm with provable convergence guarantees to solve it. Extensive experiments on various datasets demonstrate the superiority of the proposed model. MATLAB code is available at https://github.com/wangzhi-swu/NRDN-MvSC.

## 1. Introduction

Multiview data, stemming from diverse sources, is ubiquitous in real-world applications and often exhibits high dimensionality and large-scale complexity (Deng et al., 2022; Huang et al., 2021). Such inherent properties pose new challenges to efficient data analysis. To this end, multiview subspace clustering (MvSC) methods (Zhang et al., 2020; Li et al., 2022b; Yu et al., 2025a) have been devel-

oped, which aim to partition the data into multiple clusters, where each cluster corresponds to an underlying subspace. Recently, MvSC has garnered considerable attention from researchers across data mining and computer vision, and has been successfully employed in various tasks, including face clustering (Xie et al., 2018; Tang et al., 2023), outlier detection (Zhao et al., 2018; Hou et al., 2022), and motion segmentation (Rao et al., 2010; Vidal et al., 2008).

Roughly, existing MvSC approaches can be classified into two categories, namely matrix-oriented methods (Xia et al., 2014; Cao et al., 2015; Zhang et al., 2020; Zhou et al., 2020; Wang et al., 2024) and tensor-oriented methods (Wang et al., 2023a; Ji & Feng, 2023; Fu et al., 2023; Cheng et al., 2019; Fu et al., 2026). Matrix-oriented methods typically focus on analyzing the intrinsic structure within single views or the dependencies across view pairs. Despite their success, these methods often fail to fully exploit the high-order correlations across all views to derive a unified consensus structure (Wang et al., 2025a; Fu et al., 2026).

Fueled by the rapid progress of tensor algebra, which effectively exploits high-order information from a multi-dimensional perspective, low-rank tensor representation-based MvSC methods (Yu et al., 2025b; Cheng et al., 2019; Chen et al., 2024) have emerged, exhibiting superior performance in various applications. The pioneering works are t-SVD-MSC (Xie et al., 2018) and ETLMSC (Wu et al., 2019), which utilize the Tensor Nuclear Norm (TNN) on the rotated multiview representation tensor. However, TNN is not the best approximation of tensor rank (Wang et al., 2022). Therefore, a number of nonconvex approximations, such as WTNNM (Gao et al., 2020), WTSNM (Xia et al., 2022) and $\text{TLS}_p\text{NM}$ (Guo et al., 2023), have been introduced to more effectively characterize tensor low-rankness. Empirically, these non-convex approximation-based methods achieve better performance than convex TNN-based methods (Long et al., 2025). However, these methods suffer from several drawbacks, such as insufficient exploitation of adaptive data priors and the lack of a closed-form solution.

With the advent of deep learning, deep multiview clustering methods (Peng et al., 2020; Ji et al., 2017; Li et al., 2019; Xie et al., 2021) have witnessed a vigorous development. In (Peng et al., 2020), Peng et al. proposed a deep extension of

[1]College of Computer and Information Science, Southwest University, Chongqing, China [2]School of Artificial Intelligence, Chongqing University of Technology, Chongqing, China. Correspondence to: Zhi Wang <chiw@swu.edu.cn>.

*Proceedings of the 43rd International Conference on Machine Learning*, Seoul, South Korea. PMLR 306, 2026. Copyright 2026 by the author(s).

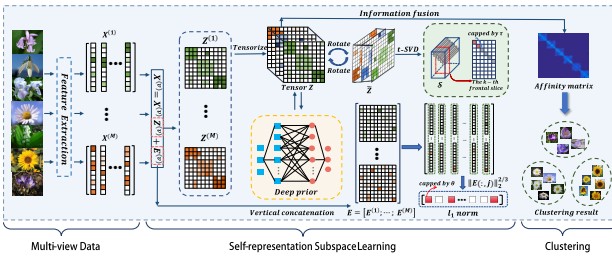

*Figure 1.* The framework of NRDN-MvSC. First, the representation matrices are explored from data features and stacked into a tensor for self-representation learning. Then, by jointly integrating nonconvex regularizers and deep networks, the proposed model can capture both underlying low-rank structure and intrinsic local structure of the data, while effectively suppressing the noise and outliers. Finally, the affinity matrix can be constructed by merging the learned representations for clustering.

*Table 1.* The summarization of notations.

| Notation | Description |
|---|---|
| $\mathcal{X}, X, \mathbf{x}, x$ | tensor, matrix, vector, scalar |
| $\mathcal{X}^{(k)}$ | the $k$-th frontal slice of the tensor $\mathcal{X}$ |
| $N, V$ | the number of samples, views |
| $d_v$ | the dimension of $v$-th view |
| $X_{i,j}$ | the $(i, j)$-th element of $X$ |
| $X^T$ | the transpose of $X$ |
| $\bar{\mathcal{X}} = \text{fft}(\mathcal{X}, [], 3)$ | the fast Fourier transform of $\mathcal{X}$ along tube fiber |
| $X^{(v)} \in \mathbb{R}^{dv \times N}$ | the feature matrix of the $v$-the view |
| $E^{(v)} \in \mathbb{R}^{dv \times N}$ | the error matrix of the $v$-the view |
| $Z^{(v)} \in \mathbb{R}^{N \times N}$ | the representation matrix of the $v$-the view |
| $\|\mathbf{x}\|_2$ | the $\ell_2$-norm of vector $\mathbf{x}$ |
| $\|\cdot\|_F, \|\cdot\|_{2,1}$ | Frobenius norm, $\ell_{2,1}$-norm |
| $\|\cdot\|_{\circledast}, \|\cdot\|_{\infty}$ | t-SVD-nuclear norm, infinity norm |

sparse subspace clustering, which simultaneously achieves data representation learning and subspace clustering. In (Ji et al., 2017), Ji et al. proposed a deep neural network architecture for subspace clustering that can directly learn the correlations between all data points. Following this line, Xie et al. in (Xie et al., 2021) suggested a joint learning strategy to leverage multiview features and complementary information. Despite demonstrating impressive performance, these deep learning-based methods typically suffer from heavy dependence on extensive training data and an inherent lack of interpretability.

To overcome above shortcomings, this work introduces a new nonconvex regularized MvSC framework with deep network, termed NRDN-MvSC, to enhance clustering performance. Compared with existing methods, the proposed approach not only more comprehensively captures the high-order correlations and complementary information inherent in multiview data, but also exhibits significantly stronger robustness to noise and outliers. More importantly, our method seamlessly integrates deep and low-rank priors, which ultimately enables superior preservation of both the global structure and the local geometric structure of the multiview data. Fig. 1 shows the overall framework of the proposed NRDN-MvSC. The novelties and main contributions of this work are outlined as follows.

- We develop a novel regularization term, referred to as capped tensor Schatten 2/3-norm (CTS$_{2/3}$N) minimization, for approximating the rank of the consensus tensor, which is designed to effectively leverage salient structural and high-order information within multiview data. Moreover, the capped $\ell_{2,2/3}$-norm is proposed to accurately characterize the error term, thereby markedly improving the robustness of our method.

- We leverage the newly proposed capped norm to develop a new tensor low-rank constrained MvSC

method. Additionally, we integrate deep networks into our method, with a focus on harnessing advanced representational capabilities while preserving model interpretability.

- Rigorous mathematical analysis of the CTS$_{2/3}$N least squares problem revealed a closed-form solution, based on which we propose an efficient optimization algorithm with provable convergence guarantees.

- Extensive experiments on diverse multiview datasets demonstrate that our method outperforms several state-of-the-art multiview clustering methods.

## 2. Notations and preliminaries

For brevity, we give the frequently used notations in Table 1. Then, the basic definitions necessary for the proposed model are provided. For tensor $\mathcal{X} \in \mathbb{R}^{n_1 \times n_2 \times n_3}$, its **transpose** is $\mathcal{X}^T \in \mathbb{R}^{n_2 \times n_1 \times n_3}$, which is obtained by transposing each frontal slice and reversing the order of transposed slices 2 to $n_3$. The tensor $\mathcal{I} \in \mathbb{R}^{n_1 \times n_1 \times n_3}$ is an **identity tensor** if its first frontal slice is an identity matrix and the rest frontal slices are zero. Now, given tensors $\mathcal{X} \in \mathbb{R}^{n_1 \times n_2 \times n_3}$ and $\mathcal{Y} \in \mathbb{R}^{n_2 \times n_4 \times n_3}$, we introduce the following related tensor operations. The block vectorization matrix is defined as $\mathbf{bvec}(\mathcal{X}) = [\mathcal{X}^{(1)}; \ldots; \mathcal{X}^{(n_3)}]$, and its inverse operation is $\mathbf{bvfold}(\mathbf{bvec}(\mathcal{X})) = \mathcal{X}$. The block circular matrix $\mathbf{bcire}(\mathcal{X})$ is constructed by

$$\mathbf{bcirc}(\mathcal{X}) = \begin{bmatrix} \mathcal{X}^{(1)} & \mathcal{X}^{(n_3)} & \ldots & \mathcal{X}^{(2)} \\ \mathcal{X}^{(2)} & \mathcal{X}^{(1)} & \ldots & \mathcal{X}^{(3)} \\ \vdots & \vdots & \ddots & \vdots \\ \mathcal{X}^{(n_3)} & \mathcal{X}^{(n_3-1)} & \ldots & \mathcal{X}^{(1)} \end{bmatrix}. \quad (1)$$

The **t-product** between $\mathcal{X}$ and $\mathcal{Y}$ is a $n_1 \times n_4 \times n_3$ tensor, which is defined as $\mathcal{X} * \mathcal{Y} = \mathbf{bvfold}\{\mathbf{bcirc}(\mathcal{X})\mathbf{bvec}(\mathcal{Y})\}$. The tensor $\mathcal{X} \in \mathbb{R}^{n_1 \times n_1 \times n_3}$ is **orthogonal** if it satisfies

$\mathcal{X}^T * \mathcal{X} = \mathcal{X} * \mathcal{X}^T = \mathcal{I}$. A tensor is f-diagonal if all its frontal slices are diagonal matrices. Based on these tensor operations, the following definitions can be defined.

**Definition 2.1** (t-SVD (Kilmer et al., 2013)). Given tensor $\mathcal{X} \in \mathbb{R}^{n_1 \times n_2 \times n_3}$, its tensor Singular Value Decomposition (t-SVD) is given by $\mathcal{X} = \mathcal{U} * \mathcal{S} * \mathcal{V}^T$, where $\mathcal{U} \in \mathbb{R}^{n_1 \times n_1 \times n_3}$ and $\mathcal{V} \in \mathbb{R}^{n_2 \times n_2 \times n_3}$ are orthogonal tensors, $\mathcal{S} \in \mathbb{R}^{n_1 \times n_2 \times n_3}$ is an $f$-diagonal tensor.

**Lemma 2.1** ((Wang et al., 2019)). *For any $\xi \geq 0$, the optimal solution to*

$$\arg\min_{\delta \geq 0} \frac{1}{2}(x - \delta)^2 + \xi x^{\frac{2}{3}}, \tag{2}$$

*is given by*

$$h_{\xi, \frac{2}{3}}(\delta) = \begin{cases} ((\phi_\xi(\delta) + \Psi_\xi(\delta))/2)^3, & if\ \delta > \frac{2}{3}(24\xi)^{\frac{1}{4}}, \\ 0, & otherwise, \end{cases} \tag{3}$$

*where $\Psi_\xi(\delta) = (2\delta/\phi_\xi(\bar{\mathcal{S}}_{i,i}^{(j)}) - (\phi_\xi(\delta))^2)^{1/2}$ with $\phi_\xi(\delta) = (32\xi)^{1/4}/\sqrt{3}(\cosh(\operatorname{arccosh}(27\delta^2(2\xi)^{-3/2}/16)/3))^{1/2}$.*

## 3. The Proposed Method

### 3.1. Problem Formulation

Typically, self-representation-based MvSC approaches utilize low-rank tensor representation learning to effectively capture high-order correlations and complementary information among multiple views. Over the past decade, the convex TNN has been widely employed for recovering low-rank tensor representations. However, TNN is generally not the tightest approximation of the tensor average rank, often resulting in suboptimal solutions. To this end, several nonconvex regularizers have been proposed to better approximate the rank function. While these nonconvex methods empirically achieve superior clustering performance compared to TNN, they typically suffer from a lack of closed-form solutions and robust theoretical guarantees.

To alleviate these problems, we first propose a new nonconvex regularizer to better approximate the tensor rank. To further enhance the model's robustness against noise and outliers, we introduce the capped $\ell_{2,2/3}$-norm regularization on the error term. They are defined as follows.

**Definition 3.1.** Given $\mathcal{Z} \in \mathbb{R}^{n_1 \times n_2 \times n_3}$, $h = \min(n_1, n_2)$, the $\text{CTS}_{2/3}\text{N}$ is defined as

$$\|\mathcal{Z}\|_{C_{S_{2/3}}^\tau}^{2/3} = \frac{1}{n_3} \sum_{j=1}^{n_3} \sum_{i=1}^{h} \min(\sigma_i(\bar{\mathcal{Z}}^{(j)}), \tau)^{2/3}, \tag{4}$$

where $\tau > 0$ is a threshold parameter.

**Definition 3.2.** The capped $\ell_{2,2/3}$-norm of $E$ is defined as

$$\|E\|_{C_{2,2/3}^\theta}^{2/3} = \sum_{j=1}^{s} \min(\|E(:,j)\|_2^{2/3}, \theta), \tag{5}$$

where $\theta \geq 0$ is a parameter.

Deep neural networks have recently demonstrated a strong capability for learning adaptive priors across a variety of machine learning tasks. Consequently, this work leverages this principle to preserve the local geometric structure of data, thereby significantly enhancing the clustering performance. The final objective function of our method is formulated as

$$\begin{aligned} &\min_{\mathcal{Z},E}\ \|\mathcal{Z}\|_{C_{S_{2/3}}^\tau}^{2/3} + \lambda_1 \|E\|_{C_{2,2/3}^\theta}^{2/3} + \lambda_2 \Theta(\mathcal{Z}) \\ &s.t.\ X^{(v)} = X^{(v)} Z^{(v)} + E^{(v)}, \\ &\quad\quad \mathcal{Z} = \Phi(Z^{(1)}, Z^{(2)}, ..., Z^{(V)}), \\ &\quad\quad E = [E^{(1)}; E^{(2)}; ...; E^{(V)}], \end{aligned} \tag{6}$$

where $\lambda_1$ and $\lambda_2$ are trade-off parameters, $\Theta(\cdot)$ is the deep network.

*Remark* 3.1 (*The superiority of* $\text{CTS}_{2/3}\text{N}$). Inspired by the Schatten capped $p$-norm (Li et al., 2022a), we develop a new approximation function $\text{CTS}_{2/3}\text{N}$. Basically, $\|\mathcal{Z}\|_{C_{S_{2/3}}^\tau}^{2/3} = 0$ when $\mathcal{Z} = 0$, and this is consistent with the rank function. When the largest singular values change significantly, the tensor rank may remain invariant, yet the TNN value will be dramatically altered. Since the $\text{CTS}_{2/3}\text{N}$ only suppresses small singular values, its value changes only slightly and may even remain unchanged. Crucially, the least-squares problem associated with $\text{CTS}_{2/3}\text{N}$ admits a closed-form solution, whereas the regularizer proposed in (Li et al., 2022a) does not.

*Remark* 3.2 (*The benefits of capped* $\ell_{2,2/3}$). Unlike conventional MvSC approaches that typically rely on the $\ell_{2,1}$ and $\ell_1$ norms, our proposed capped $\ell_{2,2/3}$ term offers superior capability in suppressing noise and filtering out outliers. The magnitude of $\ell_{2,2/3}$ is strictly bounded within $\theta$, irrespective of how far the input values deviate from zero. In practice, since outliers typically cause significant deviations from zero, this inherent boundedness grants $\ell_{2,2/3}$ superior robustness against outliers.

*Remark* 3.3 (*The benefit of low-rank and deep priors*). In Problem (6), applying low-rank and deep priors to $\mathcal{Z}$ facilitates capturing both the global and local geometric structures, which ultimately enables the effective exploitation of high-order, complementary, and consistent information across views.

### 3.2. Optimization

To solve the proposed model, an efficient and effective algorithm is proposed via using the framework of ADMM. We

introduce the auxiliary variables $\mathcal{G}$ and $\mathcal{Y}$ for the separability of $\mathcal{Z}$. Then the optimization problem (6) is equivalent to the following unconstrained form.

$$
\begin{aligned}
& \mathcal{L}(\{Z^{(v)}\}_{v=1}^V, E, \mathcal{G}, \mathcal{Y}; \{Y_1^{(v)}\}_{v=1}^V, \mathcal{Y}_2, \mathcal{Y}_3) \\
& = \|\mathcal{G}\|_{C_{S_{2/3}}^\tau}^{2/3} + \lambda_1 \|E\|_{C_{2,2/3}^\theta}^{2/3} + \lambda_2 \Theta(\mathcal{Y}) + \langle \mathcal{Y}_2, \mathcal{Z} - \mathcal{G} \rangle \\
& \quad + \frac{\mu_2}{2} \|\mathcal{Z} - \mathcal{G}\|_F^2 + \langle \mathcal{Y}_3, \mathcal{Z} - \mathcal{Y} \rangle + \frac{\mu_3}{2} \|\mathcal{Z} - \mathcal{Y}\|_F^2 \\
& \quad + \sum_{v=1}^V \Big( \langle Y_1^{(v)}, X^{(v)} - X^{(v)} Z^{(v)} - E^{(v)} \rangle \\
& \quad + \frac{\mu_1}{2} \|X^{(v)} - X^{(v)} Z^{(v)} - E^{(v)}\|_F^2 \Big)
\end{aligned}
\tag{7}
$$

where $\{Y_1^{(v)}\}_{v=1}^V$, $\mathcal{Y}_2$ and $\mathcal{Y}_3$ are Lagrangian multipliers, $\mu_1$, $\mu_2$ and $\mu_3$ denote penalty parameters to control convergence (Liu et al., 2023a). Then, each variable can be alternately updated while keeping the others fixed.

$Z^{(v)}$-**subproblem**. Fixing other variables, according to $\Phi_{(v)}^{-1}(\mathcal{Y}_2) = Y_2^{(v)}$ and $\Phi_{(v)}^{-1}(\mathcal{Y}_3) = Y_3^{(v)}$, the variable $Z^{(v)}$ can be optimized by the following subproblem

$$
\begin{aligned}
& \arg\min_{Z^{(v)}} \ \langle Y_1^{(v)}, X^{(v)} - X^{(v)} Z^{(v)} - E^{(v)} \rangle \\
& + \frac{\mu_1}{2} \|X^{(v)} - X^{(v)} Z^{(v)} - E^{(v)}\|_F^2 + \langle \mathcal{Y}_2, \mathcal{Z} - \mathcal{G} \rangle \\
& + \frac{\mu_2}{2} \|\mathcal{Z} - \mathcal{G}\|_F^2 + \langle \mathcal{Y}_3, \mathcal{Z} - \mathcal{Y} \rangle + \frac{\mu_3}{2} \|\mathcal{Z} - \mathcal{Y}\|_F^2.
\end{aligned}
\tag{8}
$$

Obviously, problem (8) is quadratic, smooth and convex, which can be solved by setting the derivative with respect to $Z^{(v)}$ to zero. Then, its solution can be obtained as

$$
\begin{aligned}
Z^{(v)} = & (\mu_2 I + \mu_3 I + \mu_1 X^{(v)^T} X^{(v)})^{-1} \\
& (\mu_1 X^{(v)^T} X^{(v)} + X^{(v)^T} Y_1^{(v)} - \mu_1 X^{(v)^T} E^{(v)} \\
& - Y_2^{(v)} - Y_3^{(v)} + \mu_2 G^{(v)} + \mu_3 Y^{(v)}).
\end{aligned}
\tag{9}
$$

$E$-**subproblem**. Fixing other variables, the variable $E$ can be optimized by the following subproblem

$$
\arg\min_E \ \frac{\lambda_1}{\mu_1} \|E\|_{C_{2,2/3}^\theta}^{2/3} + \frac{1}{2} \|E - D\|_F^2,
\tag{10}
$$

where $D$ is constructed by concatenating vertically along the column of $D^{(v)}$, i.e. $D = [D^{(1)}; D^{(2)}; \dots; D^{(V)}]$, and $D^{(v)} = X^{(v)} - X^{(v)} Z^{(v)} + \mu_1^{-1} Y_1^{(v)}$. The $j$-th column of $E$ can be obtained by

$$
E_{:,j} = \begin{cases} D_{:,j}, & \text{if } \theta \in [0, \theta^*], \\ e' \frac{D_{:,j}}{\|D_{:,j}\|_2}, & \text{if } \theta \in (\theta^*, \infty), \end{cases}
\tag{11}
$$

where $\theta^* = (\lambda_1/\mu_1)(e' - D_{:,j})^2 + (e')^{2/3}$ and $e' = h_{\frac{\lambda_1}{\mu_1}, \frac{2}{3}}(D_{:,j})$.

$\mathcal{G}$-**subproblem**. Fixing the other variables, the variable $\mathcal{G}$ can be optimized by the following subproblem

$$
\arg\min_{\mathcal{G}} \ \|\mathcal{G}\|_{C_{S_{2/3}}^\tau}^{2/3} + \frac{\mu_2}{2} \|\mathcal{G} - (\mathcal{Z} + \frac{\mathcal{Y}_2}{\mu_2})\|_F^2.
\tag{12}
$$

Problem (12) is nonconvex, which is difficult to obtain the analytical solution using traditional optimization techniques (Wang et al., 2025b; Shan et al., 2024). To this end, the following theorem[1] is proposed to address this issue.

**Theorem 3.1.** *Suppose $\mathcal{X} \in \mathbb{R}^{n_1 \times n_2 \times n_3}$ with t-SVD $\mathcal{X} = \mathcal{U} * \mathcal{S} * \mathcal{V}^T$ and $\xi > 0$. Then, the optimal solution to*

$$
\arg\min_{\mathcal{G}} \ \frac{1}{2} \|\mathcal{G} - \mathcal{X}\|_F^2 + \xi \|\mathcal{G}\|_{C_{S_{2/3}}^\tau}^{2/3}
\tag{13}
$$

*is given by*

$$
\mathcal{G}^* = \mathcal{U} * \text{ifft}(\mathcal{D}_{\xi,\tau}(\bar{\mathcal{S}}), [], 3) * \mathcal{V}^T,
\tag{14}
$$

*where $\text{ifft}(\mathcal{D}_{\xi,\tau}(\bar{\mathcal{S}}), [], 3) \in \mathbb{R}^{n_1 \times n_2 \times n_3}$ is an $f$-diagonal tensor, in which $\mathcal{D}_{\xi,\tau}(\bar{\mathcal{S}}_{i,i}^{(j)})$ satisfies the following condition*

$$
\mathcal{D}_{\xi,\tau}(\bar{\mathcal{S}}_{i,i}^{(j)}) = \arg\min_{\sigma_i \geq 0} \frac{1}{2} (\sigma_i - \bar{\mathcal{S}}_{i,i}^{(j)})^2 + \xi \min(\sigma_i, \tau)^{\frac{2}{3}}
\tag{15}
$$

*for $1 \leq j \leq n_3$ and $1 \leq i \leq \min(n_1, n_2)$.*

Effectively solving the nonconvex problem (55) is critical for developing an efficient optimization algorithm for problem (13). However, obtaining its global optimum directly remains challenging. Thus, we derive a closed-form solution for (55) as stated in the following proposition.

**Proposition 3.1.** *Given scalar $\delta \geq 0$ and $\xi \in (0, 4.5\tau^{4/3})$. If $\alpha^*$ minimize*

$$
h(\alpha) = \frac{1}{2} (\alpha - \delta)^2 + \xi \min(\alpha, \tau)^{\frac{2}{3}}, \quad s.t. \alpha \geq 0,
\tag{16}
$$

*then, $\alpha^*$ satisfies*

$$
\alpha^* = \begin{cases} \delta, & \text{if } \tau \in [0, \tau^*], \\ h_{\xi, \frac{2}{3}}(\delta), & \text{if } \tau \in (\tau^*, \infty) \end{cases}
\tag{17}
$$

*with $\tau^* = (1/(2\xi)(h_{\xi, \frac{2}{3}}(\delta) - \delta)^2 + (h_{\xi, \frac{2}{3}}(\delta))^{2/3})^{3/2}$.*

Let $\mathcal{A} = \mathcal{Z} + \mu_2^{-1} \mathcal{Y}_2$ and the t-SVD of $\mathcal{A}$ be $\mathcal{U}_A * \mathcal{S}_A * \mathcal{V}_A^T$. According to Theorem 1, the optimal solution of problem (12) can be obtained by

$$
\mathcal{G} = \mathcal{U}_A * \text{ifft}(\mathcal{D}_{\frac{1}{\mu_2}, \tau}(\bar{\mathcal{S}}_A), [], 3) * \mathcal{V}_A^T.
\tag{18}
$$

$\mathcal{Y}$-**subproblem**. Fixing the other variables and letting $\sigma = \frac{\lambda_2}{\mu_3}$, the variable $\mathcal{Y}$ can be optimized by the following subproblem

$$
\arg\min_{\mathcal{Y}} \ \Theta(\mathcal{Y}) + \frac{1}{2\sigma^2} \|\mathcal{Y} - (\mathcal{Z} + \frac{\mathcal{Y}_3}{\mu_3})\|_F^2.
\tag{19}
$$

---

[1]All the proofs can be found in the Supplementary Materials.

**Algorithm 1** The algorithm for solving problem (6).

---

**Input**: Multiview data $\{X^{(1)}, X^{(2)}, ..., X^{(V)}\}$, parameter $\lambda_1, \lambda_2, \theta, \tau$ and cluster number $K$.

**Initialized**:$Z^{(v)} = E^{(v)} = Y_1^{(v)} = \mathbf{0}, \mathcal{G} = \mathcal{Y} = \mathcal{Y}_2 = \mathcal{Y}_3 = \mathbf{0}, \mu_i = 10^{-5}, \mu_m = 10^{12}, \epsilon = 10^{-7}$.

1: **while** not converge **do**
2:    Update $\{Z^{(v)}\}_{v=1}^{V}$ by (9);
3:    Update $E$ by (11);
4:    Update $\mathcal{G}$ by (18);
5:    Update $\mathcal{Y}$ by (20);
6:    Update $\{Y_1^{(v)}\}_{v=1}^{V}, \mathcal{Y}_2, \mathcal{Y}_3, \mu_i$ by (21) - (24);
7:    Obtain $\mathcal{Z} = \Phi(Z^{(1)}, Z^{(2)}, ..., Z^{(V)})$;
8:    Check the convergence conditions:
       $\|X^{(v)} - X^{(v)}Z^{(v)} - E^{(v)}\|_\infty < \epsilon, \|\mathcal{Z} - \mathcal{G}\|_\infty < \epsilon,$
       $\|\mathcal{Z} - \mathcal{Y}\|_\infty < \epsilon;$
9: **end while**

**Output**: Optimal representation $\{Z^{(v)}\}_{v=1}^{V}$.

---

In our model, FFDNet is adopted as an implicit regularizer, which is flexible denoising convolutional neural network (Zhang et al., 2018). Feeding $\mathcal{Z} + \mu_3^{-1}\mathcal{Y}_3$ into the denoiser FFDNet, the solution of $\mathcal{Y}$-subproblem can be obtained by

$$\mathcal{Y} = \text{FFDNet}(\mathcal{Z} + \frac{\mathcal{Y}_3}{\mu_3}, \sigma). \qquad (20)$$

At last, the multipliers $Y_1^{(v)}, \mathcal{Y}_2, \mathcal{Y}_3$ and the penalty parameters $\mu_1, \mu_2, \mu_3$ can be update by

$$\begin{cases} Y_1^{(v)} = Y_1^{(v)} + \mu_1(X^{(v)} - X^{(v)}Z^{(v)} - E^{(v)}), & (21) \\ \mathcal{Y}_2 = \mathcal{Y}_2 + \mu_2(\mathcal{Z} - \mathcal{G}), & (22) \\ \mathcal{Y}_3 = \mathcal{Y}_3 + \mu_3(\mathcal{Z} - \mathcal{Y}), & (23) \\ \mu_i = \min(\eta\mu_i, \mu_m), \quad i = 1, 2, 3 & (24) \end{cases}$$

where $\eta > 1$ is used to accelerate the convergence and $\mu_m$ gives the upper bound for $\mu$. The detailed algorithm for solving our model is summarized in Algorithm 1.

### 3.3. Algorithm Analysis

**Convergence Analysis.** The convergence of the algorithm based on ADMM for non-convex problems is generally hard to justify (Liu et al., 2025b; 2023b). Here, we provide the rigorous theoretical analysis in Theorem 2, proving that the generated variable sequence converges to stable points.

**Theorem 3.2.** *The sequence generated by Algorithm 1, denoted as $\{\mathcal{T}_k = (Z_k^{(v)}, E_k^{(v)}, \mathcal{G}_k, \mathcal{Y}_k; Y_{1,k}^{(v)}, \mathcal{Y}_{2,k}, \mathcal{Y}_{3,k})\}_{k=1}^{\infty}$, adheres to the following two fundamental properties:*

*1) The sequence $\{\mathcal{T}_k\}_{k=1}^{\infty}$ is bounded.*

*2) The sequence $\{\mathcal{T}_k\}_{k=1}^{\infty}$ converges to stable points.*

*Table 2*. Statistical information for benchmark datasets.

| Type | Dataset | Dimension | Sam./Clu. | View |
|------|---------|-----------|-----------|------|
| Face | Yale | 4096/3304/6750 | 165/15 | 3 |
| | ORL | 4096/3304/6750 | 400/ 40 | 3 |
| | Notting-Hill | 2000/3304/6750 | 4660/5 | 3 |
| Digits | UCI-digits | 216/76/64 | 2000/10 | 3 |
| Text | BBC4View | 4659/4633/4665/4684 | 685/5 | 4 |
| Plant | Flower17 | 1360/$\cdots$/1360/1360 | 1360/17 | 7 |
| | 100leaves | 64/64/64 | 1660/100 | 3 |
| Object | COIL20 | 4096/3304/6750 | 1440/20 | 3 |
| | Caltech101 | 48/40/254/1984/512/928 | 9144/102 | 6 |
| Scene | MITIndoor | 1770/3600/1240/4096 | 5360/67 | 4 |

**Computational Complexity Analysis.** The computational complexity of our proposed algorithm mainly lies in updating variables $Z^{(v)}$ and $\mathcal{G}$. For updating $Z^{(v)} \in \mathbb{R}^{N \times N}$, it takes $\mathcal{O}(N^3 + N^2 d_v)$ time because it requires the operation of matrix inverse. For tensor $\mathcal{G} \in \mathbb{R}^{N \times N \times V}$, the updating rule involves SVD, FFT and inverse FFT operations, accumulating a complexity of $\mathcal{O}(V^2 N^2 + V N^2 log(N))$. Considering the fact $V \ll dv$ and $V \ll N$, the overall computational complexity of our Algorithm is $\mathcal{O}(T(N^3 + N^2 V log(N) + N^2 d_v))$ after $T$ iterations. Furthermore, the space complexity is expressed as $\mathcal{O}(VN^2 + Nd)$ in which $d = max(dv|v = 1, ..., V)$.

## 4. Experiment

### 4.1. Experiment Setup

In this section, extensive experiments are conducted on ten benchmark datasets to validate the effectiveness and superiority of our model. All experiments are performed by MATLAB R2020b on a workstation equipped with an Intel(R) Xeon(R) Gold 6230 CPU and 256GB RAM.

**Datasets.** Ten representative datasets that cover a variety of clustering scenarios are employed for the validation of our model, including Yale, ORL, Notting-Hill, UCI-digits, BBC4View, Flower17, 100leaves, COIL20, Caltech101 and MITIndoor. The general statistics can be found in Table 2.

**Baselines.** Fifteen representative clustering approaches are chosen as competitors. Twelve of them are tensor-oriented methods, including t-SVD-MSC (Xie et al., 2018), WTNNM (Gao et al., 2020), GNLTA (Chen et al., 2021), WTSNM (Xia et al., 2022), BTMSC (Wang et al., 2023b), TLS$p$NM-MSC (Guo et al., 2023), TBGL (Xia et al., 2023), L1SL-$\ell_{2,p}$-TAF-MSC (Xie et al., 2024), TC-MVSC (Chang et al., 2024), STONE (Gu & Feng, 2024), S$^2$MVTC (Long et al., 2024) and ESTMC (Ji & Feng, 2025). The NLRSC (Wang et al., 2024) is a matrix-oriented model. Furthermore, we perform two classic single-view clustering methods (LRR (Liu et al., 2013) and SSC (Elhamifar & Vidal, 2013)) on each view and report the best results.

*Table 3.* Clustering results of different methods on Yale, ORL, Notting-Hill, UCI-digits and BBC4View datasets.

| Dataset | Metric | LRR | SSC | t-SVD-MSC | WTNNM | GNLTA | WTSNM | BTMSC | TLS$p$NM-MSC | TBGL | NLSRC | L1SL | TC-MVSC | STONE | S²MVTC | ESTMC | Ours |
|---|---|---|---|---|---|---|---|---|---|---|---|---|---|---|---|---|---|
| Yale | ACC | 0.7333 | 0.5391 | 0.9643 | 0.9818 | 0.7315 | 0.9855 | 0.4909 | 0.9939 | 0.6788 | 0.7333 | 0.8727 | 0.5394 | 0.7693 | 0.8364 | 0.7212 | **1.0000** |
| | NMI | 0.7639 | 0.5615 | 0.9531 | 0.9767 | 0.7779 | 0.9839 | 0.5368 | 0.9923 | 0.6454 | 0.7606 | 0.8943 | 0.5349 | 0.5880 | 0.9216 | 0.7390 | **1.0000** |
| | AR | 0.5894 | 0.3087 | 0.9132 | 0.9590 | 0.6023 | 0.9701 | 0.3015 | 0.9865 | 0.3086 | 0.6448 | 0.7995 | 0.4558 | 0.5167 | 0.7982 | 0.5713 | **1.0000** |
| | F-Score | 0.6155 | 0.3541 | 0.9184 | 0.9613 | 0.6449 | 0.9719 | 0.3452 | 0.9873 | 0.3663 | 0.5944 | 0.8121 | 0.2323 | 0.6290 | 0.8121 | 0.5984 | **1.0000** |
| | Precision | 0.5917 | 0.3292 | 0.9109 | 0.9579 | 0.6116 | 0.9710 | 0.3368 | 0.9867 | 0.2676 | 0.6200 | 0.7924 | 0.2956 | 0.7693 | 0.7209 | 0.5787 | **1.0000** |
| | Recall | 0.6412 | 0.3830 | 0.9271 | 0.9360 | 0.6839 | 0.9728 | 0.3540 | 0.9879 | 0.5806 | 0.5971 | 0.8327 | 0.2187 | 0.6232 | 0.9297 | 0.6194 | **1.0000** |
| ORL | ACC | 0.8500 | 0.6325 | 0.9720 | 0.9725 | 0.8983 | 0.9800 | 0.9275 | **1.0000** | 0.7550 | 0.8350 | 0.9940 | 0.8000 | 0.9550 | 0.7700 | 0.8975 | **1.0000** |
| | NMI | 0.9309 | 0.7999 | 0.9945 | 0.9906 | 0.9743 | 0.9959 | 0.9777 | **1.0000** | 0.8395 | 0.9253 | 0.9987 | 0.8917 | 0.9791 | 0.9175 | 0.9493 | **1.0000** |
| | AR | 0.7869 | 0.5003 | 0.9727 | 0.9700 | 0.8950 | 0.9799 | 0.9192 | **1.0000** | 0.4013 | 0.8434 | 0.9936 | 0.5820 | 0.9384 | 0.7214 | 0.8682 | **1.0000** |
| | F-Score | 0.7920 | 0.5122 | 0.9734 | 0.9705 | 0.8975 | 0.9803 | 0.9211 | **1.0000** | 0.4215 | 0.7841 | 0.9937 | 0.5943 | 0.9398 | 0.7288 | 0.8712 | **1.0000** |
| | Precision | 0.7528 | 0.4858 | 0.9574 | 0.9471 | 0.8545 | 0.9681 | 0.9814 | **1.0000** | 0.2812 | 0.7893 | 0.9894 | 0.8475 | 0.9650 | 0.6230 | 0.8493 | **1.0000** |
| | Recall | 0.8356 | 0.5417 | 0.9900 | 0.9950 | 0.9461 | 0.9931 | 0.9528 | **1.0000** | 0.8417 | 0.7410 | 0.9982 | 0.8561 | 0.9544 | 0.8778 | 0.8944 | **1.0000** |
| Notting-Hill | ACC | 0.9056 | 0.8405 | 0.9555 | 0.9843 | 0.8558 | 0.9946 | 0.9193 | 0.9908 | 0.8496 | 0.8978 | 0.9858 | 0.8852 | 0.9625 | 0.9689 | 0.9150 | **0.9983** |
| | NMI | 0.7829 | 0.8067 | 0.8949 | 0.9558 | 0.8222 | 0.9799 | 0.8833 | 0.9657 | 0.8431 | 0.7852 | 0.9549 | 0.8032 | 0.9364 | 0.9231 | 0.8243 | **0.9924** |
| | AR | 0.7907 | 0.7879 | 0.8942 | 0.9661 | 0.8063 | 0.9878 | 0.8473 | 0.9779 | 0.8604 | 0.8170 | 0.9671 | 0.7754 | 0.9443 | 0.9361 | 0.8317 | **0.9959** |
| | F-score | 0.8359 | 0.8353 | 0.9166 | 0.9734 | 0.8497 | 0.9905 | 0.8802 | 0.9827 | 0.8928 | 0.7863 | 0.9743 | 0.8291 | 0.9567 | 0.9502 | 0.8682 | **0.9968** |
| | Precision | 0.8499 | 0.8201 | 0.9344 | 0.9842 | 0.8332 | 0.9924 | 0.8969 | 0.9874 | 0.8466 | 0.8323 | 0.9808 | 0.8852 | 0.9624 | 0.9516 | 0.8813 | **0.9976** |
| | Recall | 0.8223 | 0.8509 | 0.9002 | 0.9680 | 0.8668 | 0.9885 | 0.8641 | 0.9780 | 0.9442 | 0.8482 | 0.9678 | 0.9090 | 0.9683 | 0.9488 | 0.8556 | **0.9960** |
| UCI-digits | ACC | 0.8455 | 0.9415 | 0.9985 | 0.9975 | 0.9835 | 0.9970 | 0.9995 | 0.9990 | 0.8530 | 0.9450 | 0.7525 | 0.9190 | 0.9985 | 0.9975 | 0.9970 | **1.0000** |
| | NMI | 0.7957 | 0.8868 | 0.9962 | 0.9932 | 0.9810 | 0.9924 | 0.9986 | 0.9973 | 0.8852 | 0.8887 | 0.5510 | 0.8576 | 0.9959 | 0.9944 | 0.9918 | **1.0000** |
| | AR | 0.7409 | 0.8754 | 0.9967 | 0.9950 | 0.9883 | 0.9933 | 0.9988 | 0.9977 | 0.8299 | 0.8958 | 0.5302 | 0.8164 | 0.9967 | 0.9945 | 0.9933 | **1.0000** |
| | F-score | 0.7668 | 0.8878 | 0.9970 | 0.9950 | 0.9830 | 0.9940 | 0.9990 | 0.9980 | 0.8481 | 0.8829 | 0.5773 | 0.8351 | 0.9970 | 0.9950 | 0.9940 | **1.0000** |
| | Precision | 0.7629 | 0.8861 | 0.9970 | 0.9975 | 0.9810 | 0.9940 | 0.9990 | 0.9980 | 0.7890 | 0.8946 | 0.5742 | 0.9190 | 0.9985 | 0.9950 | 0.9940 | **1.0000** |
| | Recall | 0.7708 | 0.8896 | 0.9970 | 0.9950 | 0.9774 | 0.9940 | 0.9990 | 0.9980 | 0.9168 | 0.8934 | 0.5803 | 0.8525 | 0.9970 | 0.9951 | 0.9940 | **1.0000** |
| BBC4View | ACC | 0.9168 | 0.4715 | 0.8584 | 0.9781 | 0.9854 | 0.9299 | 0.9912 | 0.9927 | 0.3533 | 0.9197 | 0.9782 | 0.7810 | 0.8453 | 0.8934 | 0.9752 | **0.9985** |
| | NMI | 0.7699 | 0.4146 | 0.6852 | 0.9382 | 0.9546 | 0.8727 | 0.9685 | 0.9707 | 0.0303 | 0.7830 | 0.9410 | 0.5647 | 0.6520 | 0.9093 | 0.9260 | **0.9945** |
| | AR | 0.8192 | 0.2561 | 0.7250 | 0.9489 | 0.9660 | 0.8389 | 0.9793 | 0.9818 | 0.0181 | 0.8610 | 0.9563 | 0.5541 | 0.6431 | 0.9173 | 0.9447 | **0.9971** |
| | F-score | 0.8613 | 0.4466 | 0.7889 | 0.9607 | 0.9739 | 0.8439 | 0.9841 | 0.9861 | 0.3791 | 0.8236 | 0.9762 | 0.6720 | 0.7224 | 0.9377 | 0.9575 | **0.9978** |
| | Precision | 0.8680 | 0.4092 | 0.8001 | 0.9773 | 0.9829 | 0.8788 | 0.9854 | 0.9874 | 0.2380 | 0.8648 | 0.9418 | 0.7810 | 0.8452 | 0.8910 | 0.9677 | **0.9987** |
| | Recall | 0.8547 | 0.4915 | 0.7782 | 0.9446 | 0.9651 | 0.9227 | 0.9828 | 0.9848 | 0.9311 | 0.8686 | 0.9461 | 0.7701 | 0.6858 | 0.9896 | 0.9475 | **0.9969** |

**Evaluation Metrics.** Six standard evaluation metrics are adopted in our experiments, including accuracy (ACC), normalized mutual information (NMI), adjusted rank index (AR), F-score, Precision and Recall. The larger scores indicate the higher clustering quality for all metrics.

**Parameter Setting.** For the baselines, the parameters are tuned according to the instructions presented in the corresponding paper, and the optimal results are reported. For our proposed model, there are four parameters that necessitate adjustment. Specifically, the parameters $\lambda_1$, $\lambda_2$ and $\tau$ are tuned within the range $\{10^{-7}, 10^{-6}, \cdots, 10^{-1}\}$ and the search range for $\theta$ is $\{10^{-3}, 10^{-2}, \cdots, 10^{2}, 10^{3}\}$.

### 4.2. Experimental Results

The clustering results compared with fifteen clustering methods on ten datasets are shown in Tables 3 - Tables 6, in which the best and second best scores are highlighted in **bold** and underlined, respectively. Based on the presented results, we draw the following important conclusions.

- Compared with single-view clustering methods, multiview clustering methods generally exhibit superior performance, which is primarily attributed to their capability to leverage diverse information among multiple views. By exploiting consistent and complementary information embedded in multiview data, the clustering performance of these methods can be improved substantially. In addition, tensor-oriented clustering methods outperform matrix-oriented ones. The main reason is that tensor-oriented methods well exploit the spatial structure information and complementary information across different views, which is lacking in matrix-oriented methods.

- It can be seen that nonconvex low-rank regularized methods are always superior to convex ones, especially when compared with the representative baseline t-SVD-MSC. For example, on the BBC4View dataset, TLS$p$NM-MSC improves around 13.43% in terms of ACC over t-SVD-MSC. This superiority can be attributed to the ability of nonconvex regularizers to consider the intrinsic disparities among singular values, thereby providing a more accurate estimation of discriminative low-rank tensor structures and effectively exploiting the complementary information across multiple views.

- The proposed method consistently surpasses other comparison methods over all datasets. Especially, our method achieves ideal clustering performance on Yale, ORL, UCI-digits and COIL20 datasets. Furthermore, the improvement of our method on some challenging datasets is remarkable. For instance, on the MITIndoor dataset, our method outperforms the second-best method across performance metrics ACC, AR and Precision with improvement of 1.32%, 1.13% and 1.59%, respectively. These results validate that our method always obtains satisfactory clustering performance in diverse clustering applications, further verifying its robustness in complex scenarios.

- To further evaluate the robustness of the proposed method, we additionally conducted experiments by corrupting the different datasets with salt-and-pepper noise at densities of 10%, 20%, and 30%. As shown

*Table 4.* Clustering results of different methods on Flower17, 100leaves, COIL20, Caltech101 and MITIndoor datasets.

| Dataset | Metric | LRR | SSC | t-SVD-MSC | WTNNM | GNLTA | WTSNM | BTMSC | TLSpNM-MSC | TBGL | NLSRC | L1SL | TC-MVSC | STONE | S²MVTC | ESTMC | Ours |
|---|---|---|---|---|---|---|---|---|---|---|---|---|---|---|---|---|---|
| Flower17 | ACC | 0.5375 | 0.2838 | 0.9169 | 0.9250 | 0.8551 | 0.9699 | 0.9522 | 0.9640 | 0.2456 | 0.5309 | 0.6625 | 0.5368 | 0.8287 | 0.7978 | 0.9449 | **0.9765** |
| | NMI | 0.5400 | 0.2790 | 0.8959 | 0.9036 | 0.8637 | 0.9482 | 0.9383 | 0.9431 | 0.2427 | 0.5373 | 0.5288 | 0.5220 | 0.7840 | 0.8784 | 0.9305 | **0.9641** |
| | AR | 0.3693 | 0.1204 | 0.8489 | 0.8574 | 0.7644 | 0.9370 | 0.9027 | 0.9259 | 0.0464 | 0.4107 | 0.4308 | 0.2136 | 0.7092 | 0.7243 | 0.8904 | **0.9513** |
| | F-Score | 0.4064 | 0.1720 | 0.8577 | 0.8658 | 0.7783 | 0.9407 | 0.9083 | 0.9303 | 0.1474 | 0.3684 | 0.4641 | 0.2769 | 0.7263 | 0.7426 | 0.8968 | **0.9541** |
| | Precision | 0.4010 | 0.1703 | 0.8556 | 0.8610 | 0.7676 | 0.9401 | 0.9051 | 0.9290 | 0.0810 | 0.4056 | 0.4611 | 0.5471 | 0.8087 | 0.6511 | 0.8929 | **0.9538** |
| | Recall | 0.4120 | 0.1737 | 0.8597 | **0.9707** | 0.7892 | 0.9412 | 0.9117 | 0.9315 | 0.8281 | 0.4005 | 0.4670 | 0.4500 | 0.7325 | 0.8640 | 0.9007 | 0.9545 |
| 100leaves | ACC | 0.7963 | 0.8275 | 0.9419 | 0.9369 | 0.9206 | 0.9443 | 0.9056 | 0.9456 | 0.6781 | 0.8175 | 0.9612 | 0.5244 | 0.8856 | 0.4400 | 0.8725 | **0.9631** |
| | NMI | 0.8925 | 0.9183 | 0.9847 | 0.9839 | 0.9763 | 0.9867 | 0.9866 | 0.9810 | 0.7324 | 0.9023 | 0.9921 | 0.6355 | 0.9688 | 0.8531 | 0.9551 | **0.9922** |
| | AR | 0.7057 | 0.7593 | 0.9352 | 0.9374 | 0.9101 | 0.9409 | 0.9104 | 0.9476 | 0.1211 | 0.7531 | 0.9598 | 0.0701 | 0.8847 | 0.4441 | 0.8488 | **0.9604** |
| | F-Score | 0.7085 | 0.7617 | 0.9359 | 0.9380 | 0.9110 | 0.9414 | 0.9113 | 0.9481 | 0.1360 | 0.7268 | 0.9601 | 0.0864 | 0.8859 | 0.4519 | 0.8502 | **0.9608** |
| | Precision | 0.6903 | 0.7357 | 0.9110 | 0.8996 | 0.8813 | 0.9139 | 0.8724 | 0.9187 | 0.1748 | 0.7295 | 0.9354 | 0.0462 | 0.9144 | 0.3011 | 0.8012 | **0.9364** |
| | Recall | 0.7277 | 0.7895 | 0.9622 | 0.9798 | 0.9428 | 0.9708 | 0.9539 | 0.9794 | 0.7400 | 0.7073 | 0.9863 | 0.6638 | 0.9358 | 0.9053 | 0.9056 | **0.9865** |
| COIL20 | ACC | 0.7792 | 0.7792 | 0.7711 | 0.8347 | 0.8496 | 0.9940 | 0.9703 | 0.9576 | 0.8104 | 0.7893 | 0.9729 | 0.7118 | 0.9229 | 0.7535 | 0.9264 | **1.0000** |
| | NMI | 0.8468 | 0.8326 | 0.8587 | 0.8967 | 0.8997 | 0.9987 | 0.9682 | 0.9576 | 0.9183 | 0.8574 | 0.9791 | 0.7848 | 0.9535 | 0.8828 | 0.9199 | **1.0000** |
| | AR | 0.7047 | 0.7015 | 0.7244 | 0.8143 | 0.8156 | 0.9936 | 0.9439 | 0.9281 | 0.7933 | 0.7742 | 0.9528 | 0.5754 | 0.8953 | 0.6815 | 0.8596 | **1.0000** |
| | F-Score | 0.7198 | 0.7168 | 0.7384 | 0.8235 | 0.8247 | 0.9937 | 0.9470 | 0.9281 | 0.8050 | 0.7403 | 0.9552 | 0.5993 | 0.9005 | 0.7003 | 0.8665 | **1.0000** |
| | Precision | 0.6933 | 0.6924 | 0.7168 | 0.8211 | 0.8205 | 0.9894 | 0.9404 | 0.9246 | 0.7006 | 0.7530 | 0.9461 | 0.5238 | 0.9229 | 0.5846 | 0.8658 | **1.0000** |
| | Recall | 0.7485 | 0.7430 | 0.7615 | 0.8258 | 0.8290 | 0.9982 | 0.9512 | 0.9388 | 0.9459 | 0.7329 | 0.9644 | 0.7002 | 0.9057 | 0.8733 | 0.8672 | **1.0000** |
| Caltech101 | ACC | 0.1032 | 0.1954 | 0.4739 | 0.6007 | 0.6408 | 0.6072 | 0.6572 | 0.6547 | 0.2002 | 0.2807 | 0.6697 | 0.2008 | 0.6511 | 0.4452 | 0.6490 | **0.6733** |
| | NMI | 0.2904 | 0.4044 | 0.7126 | 0.7984 | 0.8829 | 0.8775 | **0.9029** | 0.8547 | 0.1876 | 0.5006 | 0.9013 | 0.2536 | 0.8827 | 0.7710 | 0.8519 | 0.8985 |
| | AR | 0.0551 | 0.1347 | 0.3165 | 0.4316 | 0.4953 | 0.4415 | 0.4966 | 0.4992 | 0.0055 | 0.1498 | 0.5033 | 0.3025 | 0.4948 | 0.4962 | 0.4849 | **0.5258** |
| | F-score | 0.0318 | 0.1501 | 0.3284 | 0.4408 | 0.5040 | 0.4502 | 0.5051 | 0.5076 | 0.0309 | 0.1960 | 0.5082 | 0.0524 | 0.5035 | 0.5106 | 0.4938 | **0.5340** |
| | Precision | 0.0320 | 0.4034 | 0.6876 | 0.7647 | 0.8260 | 0.7910 | 0.8377 | 0.8649 | 0.2451 | 0.2096 | 0.8224 | 0.0280 | 0.8263 | 0.5059 | 0.8266 | **0.8698** |
| | Recall | 0.0612 | 0.1655 | 0.3129 | 0.3096 | 0.3626 | 0.3616 | 0.3727 | 0.3749 | 0.0297 | 0.3489 | 0.3675 | 0.4030 | 0.3620 | **0.5153** | 0.3590 | 0.3853 |
| MITIndoor | ACC | 0.4326 | 0.3569 | 0.6839 | 0.7661 | 0.8063 | 0.9295 | 0.9547 | 0.9405 | 0.0424 | 0.4487 | 0.9601 | 0.1313 | 0.7687 | 0.3545 | 0.8968 | **0.9733** |
| | NMI | 0.5492 | 0.4827 | 0.7502 | 0.8492 | 0.9152 | 0.9823 | 0.9718 | 0.9669 | 0.0356 | 0.5629 | 0.9883 | 0.1592 | 0.8893 | 0.7186 | 0.9473 | **0.9896** |
| | AR | 0.2899 | 0.2076 | 0.5554 | 0.6908 | 0.7619 | 0.9475 | 0.9249 | 0.9248 | 0.0007 | 0.3230 | 0.9572 | 0.0044 | 0.7301 | 0.3379 | 0.8683 | **0.9685** |
| | F-score | 0.3007 | 0.2195 | 0.5623 | 0.7061 | 0.7664 | 0.9483 | 0.9141 | 0.9087 | 0.0297 | 0.3040 | 0.9578 | 0.0330 | 0.7342 | 0.3527 | 0.8702 | **0.9690** |
| | Precision | 0.2904 | 0.2143 | 0.5430 | 0.6854 | 0.7462 | 0.9274 | 0.8879 | 0.9415 | 0.0151 | 0.3145 | 0.9355 | 0.1340 | 0.8015 | 0.2339 | 0.8499 | **0.9574** |
| | Recall | 0.3118 | 0.225 | 0.5822 | 0.7220 | 0.7876 | 0.9701 | 0.9419 | 0.9236 | 0.9477 | 0.3064 | 0.9801 | 0.6452 | 0.7686 | 0.7162 | 0.8916 | **0.9809** |

*Table 5.* Clustering results of different methods on the Yale, ORL and 100leaves datasets under various noise levels.

| Dataset | Noise | t-SVD-MSC | | WTNNM | | GNLTA | | WTSNM | | TLSpNM | | L1SL | | ESTMC | | Ours | |
|---|---|---|---|---|---|---|---|---|---|---|---|---|---|---|---|---|---|
| | | ACC | NMI | ACC | NMI | ACC | NMI | ACC | NMI | ACC | NMI | ACC | NMI | ACC | NMI | ACC | NMI |
| Yale | 10% | 0.7576 | 0.8140 | 0.7758 | 0.8335 | 0.6848 | 0.7312 | 0.9515 | 0.9550 | 0.9091 | 0.9055 | 0.4000 | 0.4346 | 0.6364 | 0.6642 | **0.9939** | **0.9924** |
| | 20% | 0.6061 | 0.7565 | 0.6727 | 0.7613 | 0.5758 | 0.6025 | 0.8000 | 0.8388 | 0.7879 | 0.8459 | 0.3698 | 0.4016 | 0.5879 | 0.6000 | **0.8601** | **0.8601** |
| | 30% | 0.4970 | 0.6450 | 0.4242 | 0.5445 | 0.4061 | 0.4457 | 0.4485 | 0.5726 | 0.5091 | 0.5552 | 0.3697 | 0.3801 | 0.3030 | 0.3239 | **0.7636** | **0.8152** |
| ORL | 10% | 0.9650 | 0.9930 | 0.9700 | 0.9936 | 0.7763 | 0.8913 | 0.9600 | 0.9867 | 0.9425 | 0.9750 | 0.6500 | 0.7555 | 0.8975 | 0.9599 | **0.9950** | **0.9955** |
| | 20% | 0.9225 | 0.9696 | 0.9300 | 0.9741 | 0.7165 | 0.8673 | 0.9450 | 0.9745 | 0.9075 | 0.9553 | 0.2775 | 0.4913 | 0.6625 | 0.7865 | **0.9500** | **0.9766** |
| | 30% | 0.7700 | 0.8850 | 0.7525 | 0.8745 | 0.6895 | 0.8551 | 0.9275 | 0.9741 | 0.7750 | 0.8847 | 0.2700 | 0.4866 | 0.2775 | 0.4854 | **0.9375** | **0.9487** |
| 100leaves | 10% | 0.1944 | 0.4782 | 0.1238 | 0.4226 | 0.7148 | 0.8917 | 0.1944 | 0.4779 | 0.4688 | 0.6731 | 0.2706 | 0.5266 | 0.8300 | 0.9261 | **0.8881** | **0.9638** |
| | 20% | 0.1219 | 0.4274 | 0.1194 | 0.4274 | 0.6909 | 0.8833 | 0.1638 | 0.4639 | 0.3856 | 0.6198 | 0.2381 | 0.5032 | 0.7488 | 0.8990 | **0.8363** | **0.9290** |
| | 30% | 0.1181 | 0.4270 | 0.1125 | 0.4205 | 0.6455 | 0.8568 | 0.1288 | 0.4379 | 0.1762 | 0.4666 | 0.2906 | 0.5421 | 0.1038 | 0.4203 | **0.7838** | **0.9129** |

in Table 5, the proposed method consistently outperforms all other compared methods across all datasets in terms of ACC and NMI. These results indicate that the proposed method achieves highly robust clustering performance in challenging noisy environments. This success stems from its simultaneous extraction of the global low-rank structure and suppression of outliers, coupled with the deep network's ability to preserve local structural properties.

- Most tensor-oriented approaches strongly depend on the ordered structure of data, which limits their capability in handling shuffled samples. To validate the effectiveness of our method under shuffled settings, we conducted comparisons with six classic t-SVD-based methods and two methods (LMTC (Liu et al., 2025a) and SE-FTMC (Ji & Feng, 2025)) which are specifically designed for shuffled scenarios. As shown in Table 6, our method remains competitive and consistently outperforms most methods while the performance decreases without the pre-sorted structural prior. Notably, it still surpasses the state-of-the-art LMTC in both ACC and NMI, which demonstrates that our mathematical

design maintains superior robustness and representation capability compared to existing alternatives.

### 4.3. Qualitative Study

**Visualization of Embedded Representation.** To visually investigate the superiority of the proposed method, the embedding results generated by our method and several comparison methods on Yale and COIL20 datasets are exhibited in Fig. 2, in which different colors denote distinct clusters. It can be intuitively observed that the samples in Fig. 2(a) - Fig. 2(c) are scattered and disordered, making it difficult to classify correctly. In contrast, our method yields clearly separable clusters, characterized by discernible margins between different groups and high compactness within each cluster. These findings strongly validate that our method can recover coherent patterns with a compact cluster structure, thereby achieving excellent clustering performance.

**Confusion Matrices.** To further verify the effectiveness of our method, the confusion matrices for clustering results obtained by three methods on the BBC4View and Yale datasets are visualized. As illustrated in Fig. 3, the row and column

*Table 6.* Clustering results of different methods on the fully shuffled Yale, ORL, COIL20, and UCI-digits datasets.

| Dataset | t-SVD-MSC | | WTNNM | | WTSNM | | TLSpNM | | BSTMC | | ESTMC | | LMTC | | SE-FTMC | | Ours | |
|---|---|---|---|---|---|---|---|---|---|---|---|---|---|---|---|---|---|---|
| | ACC | NMI | ACC | NMI | ACC | NMI | ACC | NMI | ACC | NMI | ACC | NMI | ACC | NMI | ACC | NMI | ACC | NMI |
| Yale | 0.7285 | 0.7522 | 0.6485 | 0.7152 | 0.6545 | 0.6547 | 0.5939 | 0.6605 | 0.2364 | 0.2596 | 0.6970 | 0.7075 | 0.7394 | 0.7521 | **0.7576** | **0.7828** | **0.7576** | 0.7805 |
| ORL | 0.8425 | 0.9297 | 0.8125 | 0.8990 | 0.7725 | 0.8879 | 0.8150 | 0.9205 | 0.7325 | 0.8479 | 0.8075 | 0.9024 | 0.8297 | 0.9160 | 0.8250 | 0.9149 | **0.8450** | **0.9332** |
| COIL20 | 0.7813 | 0.8422 | 0.7563 | 0.8259 | 0.7569 | 0.7917 | 0.7646 | 0.8345 | 0.7958 | 0.8441 | 0.7576 | 0.831 | 0.7569 | 0.8379 | 0.7472 | 0.8294 | **0.8201** | **0.8693** |
| UCI | 0.8785 | 0.8058 | 0.9305 | 0.8599 | 0.776 | 0.7002 | 0.9265 | 0.8593 | 0.9150 | 0.8406 | 0.8765 | 0.779 | 0.9345 | 0.8652 | 0.9235 | 0.8669 | **0.9355** | **0.8782** |

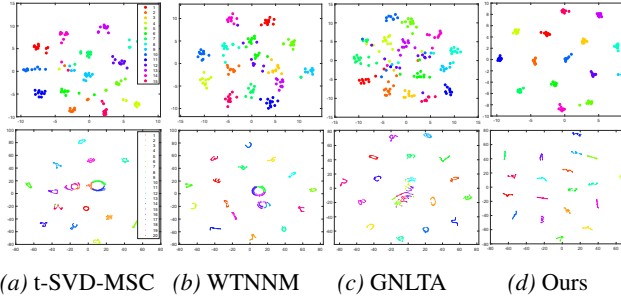

*(a) t-SVD-MSC    (b) WTNNM    (c) GNLTA    (d) Ours*

*Figure 2.* Visualization results of different methods on Yale dataset (the first row) and COIL20 dataset (the second row).

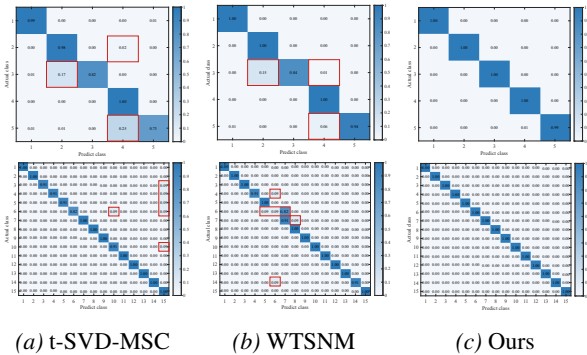

*(a) t-SVD-MSC         (b) WTSNM         (c) Ours*

*Figure 3.* The confusion matrices comparison on BBC4View (the first row) and Yale datasets (the second row).

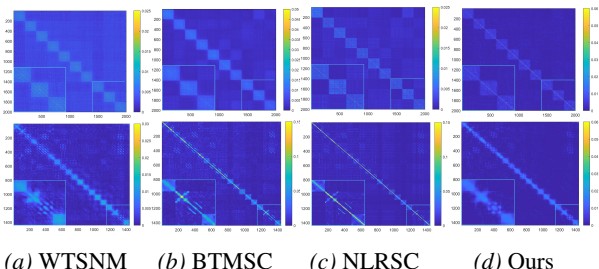

*(a) WTSNM    (b) BTMSC    (c) NLRSC    (d) Ours*

*Figure 4.* Comparison of affinity matrices of four methods on UCI-digits (the first row) and COIL20 datasets (the second row).

*Table 7.* Ablation study of the proposed model.

| Datasets | $T_1$ | $T_2$ | ACC | NMI | AR | F-score | Recall |
|---|---|---|---|---|---|---|---|
| Yale | ✗ | ✗ | 0.9643 | 0.9531 | 0.9132 | 0.9184 | 0.9271 |
| | ✓ | ✗ | 0.9942 | 0.9924 | 0.9863 | 0.9870 | 0.9879 |
| | ✓ | ✓ | **1.0000** | **1.0000** | **1.0000** | **1.0000** | **1.0000** |
| COIL20 | ✗ | ✗ | 0.7711 | 0.8587 | 0.7244 | 0.7384 | 0.7615 |
| | ✓ | ✗ | 0.9730 | 0.9740 | 0.9480 | 0.0500 | 0.9560 |
| | ✓ | ✓ | **1.0000** | **1.0000** | **1.0000** | **1.0000** | **1.0000** |
| BBC4View | ✗ | ✗ | 0.8584 | 0.6852 | 0.7250 | 0.7889 | 0.7782 |
| | ✓ | ✗ | 0.9825 | 0.9547 | 0.9591 | 0.9685 | 0.9552 |
| | ✓ | ✓ | **0.9985** | **0.9945** | **0.9971** | **0.9978** | **0.9969** |
| Flower17 | ✗ | ✗ | 0.9169 | 0.8959 | 0.8489 | 0.8577 | 0.8597 |
| | ✓ | ✗ | 0.9654 | 0.9454 | 0.9287 | 0.8329 | 0.9341 |
| | ✓ | ✓ | **0.9765** | **0.9641** | **0.9513** | **0.9541** | **0.9545** |
| 100leaves | ✗ | ✗ | 0.9419 | 0.9847 | 0.9352 | 0.9359 | 0.9622 |
| | ✓ | ✗ | 0.9581 | 0.9904 | 0.9562 | 0.9566 | 0.9802 |
| | ✓ | ✓ | **0.9631** | **0.9922** | **0.9604** | **0.9608** | **0.9865** |

correspond to the ground-truth and predicted classes, respectively. One can see that the confusion matrices in Fig. 3(a) and Fig. 3(b) exhibit noticeable misclassification in several categories, leading to degraded clustering accuracy. In contrast, our method exhibits superior performance in Fig. 3(c) since it achieves perfect diagonal dominance, indicating no confusion among different clusters. This advantage is due to the ability of our learned representations to effectively encode the cluster structure.

**Affinity Matrices.** Furthermore, the affinity graphs constructed by different methods on the UCI-Digits and COIL20 datasets are visualized in Fig. 4. It can be intuitively seen that the affinity matrices in Fig. 4(a) - Fig. 4(c) exhibit ambiguous block diagonal structures and contain more deceptive pairwise correlations, while our method yields a more prominent block-diagonal structure. It indicates that our method can reveal more discriminative subspace information and characterize the cluster structure well,

thus leading to superior clustering performance.

### 4.4. Ablation Study

Comprehensive ablation experiments are conducted on five datasets to systematically evaluate the contributions of different modules within our model. Specifically, three configurations are considered for comparison. In case I, the t-SVD-MSC model based on tensor nuclear norm serves as the baseline. In case II, we incorporate the $\mathrm{CTS}_{2/3}\mathrm{N}$ and the capped $\ell_{2,2/3}$-norm as the low-rank regularizer and error term, respectively, denoted as $T_1$. In case III, the deep prior module, denoted as $T_2$, is further integrated into Case II, forming our complete proposed model (6). The quantitative evaluations across five metrics are presented in Table 7. We can see that the clustering performance of the degraded models, achieved by t-SVD-MSC and the variant without the deep prior module, is notably inferior to the complete model (6). This clearly demonstrates the successful collaboration of $\mathrm{CTS}_{2/3}\mathrm{N}$, capped $\ell_{2,2/3}$-norm, and deep prior within the model framework, allowing them to synergistically leverage the abundant information embedded in multiview data and

*Table 8.* Clustering results of different regularizers on the Yale, ORL and 100leaves datasets under various noise levels.

| Dataset | Noise | t-SVD-MSC | | WTNNM | | GNLTA | | WTSNM | | TLSpNM | | TC-MVSC | | STONE | | ESTMC | | Ours | |
|---|---|---|---|---|---|---|---|---|---|---|---|---|---|---|---|---|---|---|---|
| | | ACC | NMI | ACC | NMI | ACC | NMI | ACC | NMI | ACC | NMI | ACC | NMI | ACC | NMI | ACC | NMI | ACC | NMI |
| Yale | 0% | 0.9643 | 0.9531 | 0.9818 | 0.9767 | 0.9636 | 0.9578 | 0.9818 | 0.9773 | 0.9939 | 0.9923 | 0.9879 | 0.9848 | 0.9939 | 0.9923 | 0.9939 | 0.9923 | **0.9942** | **0.9924** |
| | 20% | 0.6061 | 0.7565 | 0.6727 | 0.7613 | 0.8182 | 0.8785 | 0.7818 | 0.8376 | 0.8121 | 0.8694 | 0.4364 | 0.5932 | 0.7576 | 0.8458 | 0.7273 | 0.7712 | **0.8364** | **0.8634** |
| | 30% | 0.4970 | 0.6450 | 0.4242 | 0.5445 | 0.6788 | 0.7101 | 0.6909 | 0.7280 | 0.5212 | 0.6078 | 0.4182 | 0.5563 | 0.6727 | 0.7511 | 0.5939 | 0.7533 | **0.7394** | **0.7701** |
| ORL | 0% | 0.9720 | 0.9945 | 0.9725 | 0.9906 | **1.0000** | **1.0000** | 0.9650 | 0.9930 | **1.0000** | **1.0000** | **1.0000** | **1.0000** | **1.0000** | **1.0000** | 0.9725 | 0.9942 | **1.0000** | **1.0000** |
| | 20% | 0.9225 | 0.9696 | 0.9300 | 0.9741 | **0.9600** | **0.9847** | 0.9325 | 0.9662 | 0.8925 | 0.9462 | 0.8850 | 0.9287 | 0.9275 | 0.9829 | 0.9350 | 0.9659 | **0.9600** | 0.9845 |
| | 30% | 0.7700 | 0.8850 | 0.7525 | 0.8745 | 0.8775 | 0.9352 | 0.8625 | 0.9358 | 0.7950 | 0.8972 | 0.7900 | 0.8908 | 0.7925 | 0.8981 | 0.7925 | 0.8829 | **0.8950** | **0.9518** |
| 100leaves | 0% | 0.9419 | 0.9847 | 0.9369 | 0.9839 | 0.9375 | 0.9828 | 0.9450 | 0.9886 | 0.9556 | 0.9865 | 0.9475 | 0.9888 | 0.9356 | 0.9851 | 0.9431 | 0.9867 | **0.9563** | **0.9905** |
| | 20% | 0.1219 | 0.4274 | 0.1194 | 0.4274 | 0.1725 | 0.4625 | 0.1631 | 0.4656 | 0.3619 | 0.5960 | 0.2075 | 0.4850 | 0.6088 | 0.7788 | 0.2400 | 0.5107 | 0.5644 | 0.7445 |
| | 30% | 0.1181 | 0.4270 | 0.1125 | 0.4205 | 0.1175 | 0.4245 | 0.1406 | 0.4476 | 0.1806 | 0.4664 | 0.1569 | 0.4546 | 0.2881 | 0.5428 | 0.1669 | 0.4643 | **0.3544** | **0.6110** |

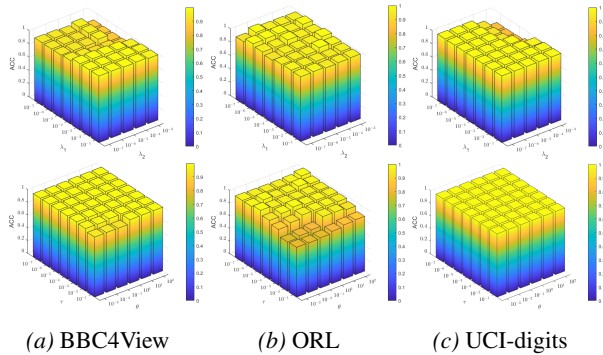

*(a) BBC4View*      *(b) ORL*      *(c) UCI-digits*

*Figure 5.* Sensitivity analysis of $\lambda_1$, $\lambda_2$, $\tau$ and $\theta$ in terms of ACC over BBC4View, ORL and UCI-digits datasets.

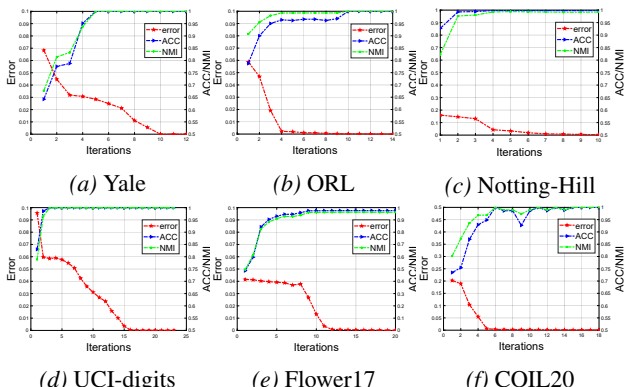

*(a) Yale*      *(b) ORL*      *(c) Notting-Hill*

*(d) UCI-digits*      *(e) Flower17*      *(f) COIL20*

*Figure 6.* Relative errors, ACC, and NMI curves on six datasets.

achieve commendable clustering performance.

Furthermore, to verify the superiority of our proposed non-convex regularizer, we consider the following basic model

$$\min_{\mathcal{Z},E} \ \Upsilon(\mathcal{Z}) + \|E\|_{2,1}, \qquad (25)$$

where $\Upsilon(\mathcal{Z})$ can be instantiated as the standard TNN, our proposed $CTS_{2/3}N$, WTNN, the Laplace function, WTSN, TLSpN, the tensor Schatten $p$-norm, HTR, or ETR. The clustering results under varying densities of salt-and-pepper noise (0%, 20%, and 30%) are reported in Table 8. It can be observed that the proposed regularizer consistently delivers competitive performance across all scenarios. For example, on the Yale dataset, our method exhibits the strongest robustness, achieving ACC scores of 0.9942, 0.8364 and 0.7394, respectively, which substantially outperform the competing nonconvex surrogates under severe corruption. These results strongly confirm that the proposed $CTS_{2/3}N$ provides a more accurate and robust approximation to the tensor rank, effectively preserving the underlying structural information of data even in heavily corrupted environments.

### 4.5. Parameter Sensitivity

To assess the sensitivity of parameters $\lambda_1$, $\lambda_2$, $\tau$ and $\theta$, an extensive grid search is conducted to identify their optimal configurations. The influence of different parameter combinations on the BBC4View, ORL, and UCI-Digits datasets is depicted in Fig. 5, highlighting fluctuations in performance according to the chosen values. As shown in Fig. 5, the

performance of our model is stable and the ACC maintains a relatively higher score within a wide range of parameters. This observation strongly illustrates the robustness and efficiency of the proposed model.

### 4.6. Convergence Analysis

To further validate the convergence stability of our method, we analyze the iterative behaviors of error, ACC, and NMI on six datasets. As illustrated in Fig. 6, the relative error decreases rapidly within 10 iterations and approaches 0 after approximately 15 iterations, while both ACC and NMI increase steadily and converge to the optimal value about 15 iterations. These observations substantiate the robust convergence property and excellent clustering performance of the proposed method.

## 5. Conclusion

In this paper, we proposed a novel NRDN-MvSC model by employing the nonconvex regularization for multi-view clustering. On one hand, it utilizes $CTS_{2/3}N$ to fully exploit the high-order correlation in multi-view data and enhance the robustness to noise and outliers. On the other hand, the global structure and the local geometric structure of multi-view data can be simultaneously preserved by introducing deep networks. Then, an efficient optimization scheme with convergence guarantee was developed to solve our model. Extensive experiments on various datasets demonstrated that our model outperforms the state-of-the-art models.

## Acknowledgements

We gratefully acknowledge the four anonymous reviewers for their valuable remarks and thoughtful suggestions. This work was supported in part by the National Natural Science Foundation of China under Grant 62372385, and in part by the High Performance Computing (HPC) clusters at Southwest University.

## Impact Statement

This paper presents work whose goal is to advance the field of Machine Learning. There are many potential societal consequences of our work, none which we feel must be specifically highlighted here.

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

# A. Proofs of theorems, lemmas and propositions

### A.1. Proofs of Theorem 1

Before proving Theorem 1, we first provide two important lemmas. Specially, we consider the schatten capped 2/3 norm minimization problem in the complex matrix case.

**Lemma A.1.** *Suppose $X$, $Y \in \mathbb{C}^{m \times n}$ are two matrices, and $Y = U_Y S V_Y^H$ is the SVD of Y. A complex matrix schatten capped 2/3 norm minimization problem is*

$$\arg\min_X \frac{1}{2}\|X - Y\|_F^2 + \xi\|X\|_{C_{S_{2/3}}^\tau}^{2/3}. \tag{26}$$

*Then, the minimizer of the matrix schatten capped 2/3 minimization is*

$$X^* = U_Y D_{\xi,\tau}(S) V_Y^H, \tag{27}$$

*where $D_{\xi,\tau}(S)$ is a diagonal matrix, its i-th diagonal element $D_{\xi,\tau}(S_{i,i})$ is given by*

$$\arg\min_{\sigma_i \geq 0} \frac{1}{2}(\sigma_i - S_{i,i})^2 + \xi min(\sigma_i, \tau)^{\frac{2}{3}} \tag{28}$$

*for $1 \leq i \leq min(m, n)$.*

Before proving Lemma A.1, we introduce Lemma A.2 as follows.

**Lemma A.2.** *(Mirsky, 1975) If complex matrix A, $B \in \mathbb{C}^{m \times n}$ have singular values $\sigma_1(A) \geq ... \geq \sigma_t(A) \geq 0$ and $\sigma_1(B) \geq ... \geq \sigma_t(B) \geq 0$, then have the following inequality.*

$$\langle A, B \rangle \leq \sum_{i=1}^t \sigma_i(A)\sigma_i(B), \tag{29}$$

*where $t = \min(m, n)$.*

The equality in (29) holds if and only if $A = UDiag(\sigma(A))V$ and $B = UDiag(\sigma(B))V$ are the SVD of A and B, respectively.

Now we begin to prove Lemma A.1.

*Proof.* Without loss of generality, we denote $\sigma : \sigma_1 \geq \cdots \geq \sigma_t \geq 0$ with $t = min(m, n)$ as the singular values of $X$. According to SVD $Y = U_Y S V_Y^H$ and Lemma A.2, we have

$$\begin{aligned}
\|X - Y\|_F^2 &= \langle X, X \rangle - 2\langle X, Y \rangle + \langle Y, Y \rangle \\
&= \sum_{i=1}^t \sigma_i^2 - 2\langle X, Y \rangle + \sum_{i=1}^t S_{i,i}^2 \\
&\geq \sum_{i=1}^t \sigma_i^2 - 2\sum_{i=1}^t \sigma_i S_{i,i} + \sum_{i=1}^t S_{i,i}^2 \\
&= \sum_{i=1}^t (\sigma_i - S_{i,i})^2.
\end{aligned} \tag{30}$$

This indicates that

$$\frac{1}{2}\|X - Y\|_F^2 + \xi\|X\|_{C_{S_{2/3}}^\tau}^{2/3} \geq \frac{1}{2}\sum_{i=1}^t (\sigma_i - S_{i,i})^2 + \xi \sum_{i=1}^t min(\sigma_i, \tau)^{\frac{2}{3}}. \tag{31}$$

The equality in (31) holds if and only if the SVD of $X$ satisfies $X = U_Y \mathrm{Diag}(\sigma_1, \cdots, \sigma_t) V_Y^H$. Hence, the minimization problem (26) can be degenerated into the following problem

$$\min_{\sigma : \sigma_1 \geq \cdots \geq \sigma_t \geq 0} \sum_{i=1}^{t} \frac{1}{2}(\sigma_i - S_{i,i})^2 + \xi \sum_{i=1}^{t} \min(\sigma_i, \tau)^{\frac{2}{3}}. \tag{32}$$

And, the optimal solution of (26) is given by

$$X^* = U_Y \mathrm{Diag}(\sigma^*) V_Y^H, \tag{33}$$

where $\sigma^*$ is the minimizer of (32).

Even if we transform the minimization problem (26) into (32), it is still difficult to find the close-form solution to the optimization problem (32) due to the order constraint and the non-convexity of the objective function. It is worth noting that if the order constraint always holds, the problem (32) is separable and can be decomposed into $t$ independent subproblems:

$$\sigma_i^* = \arg\min_{\sigma_i \geq 0} \frac{1}{2}(\sigma_i - S_{i,i})^2 + \xi \min(\sigma_i, \tau)^{\frac{2}{3}} \tag{34}$$

for $i = 1, 2, \ldots, t$. Now we can prove that the order constraint always holds for the solution of (33), i.e., $\sigma_i^* \geq \sigma_j^*$, if $i \leq j$. Applying the optimality of $\sigma_i^*, \sigma_j^*$, we have

$$(\sigma_i^* - S_{i,i})^2 + \xi \min(\sigma_i^*, \tau)^{\frac{2}{3}} \leq (\sigma_j^* - S_{i,i})^2 + \xi \min(\sigma_j^*, \tau)^{\frac{2}{3}} \tag{35}$$

and

$$(\sigma_j^* - S_{j,j})^2 + \xi \min(\sigma_j^*, \tau)^{\frac{2}{3}} \leq (\sigma_i^* - S_{j,j})^2 + \xi \min(\sigma_i^*, \tau)^{\frac{2}{3}}. \tag{36}$$

Summing (35) and (36) gives

$$(\sigma_i^* - S_{i,i})^2 + (\sigma_j^* - S_{j,j})^2 \leq (\sigma_j^* - S_{i,i})^2 + (\sigma_i^* - S_{j,j})^2. \tag{37}$$

It can reduces to

$$(\sigma_i^* - \sigma_j^*)(S_{i,i} - S_{j,j}) \geq 0. \tag{38}$$

Note that $S_{i,i} \geq S_{j,j}$ if $i \leq j$, thus we have $\sigma_i^* \geq \sigma_j^*$. Hence, the optimal solution $\sigma^*$ of (34) satisfies the order constraint.

In summary, the optimal solution of (26) is given by

$$X^* = U_Y \mathrm{Diag}(\sigma^*) V_Y^H, \tag{39}$$

where $\sigma^* = \{\sigma_1^*, \ldots, \sigma_t^*\}$ in which $\sigma_i^*$ is solving by

$$\sigma_i^* = \arg\min_{\sigma_i \geq 0} \frac{1}{2}(\sigma_i - S_{i,i})^2 + \xi \min(\sigma_i, \tau)^{\frac{2}{3}}. \tag{40}$$

$\square$

Now we are ready to prove Theorem 1.

*Proof.* In the Fourier domain, according to the fact that $\|\mathcal{A}\|_F^2 = \frac{1}{n_3}\|\bar{\mathcal{A}}\|_F^2$ with tensor $\mathcal{A} \in \mathbb{R}^{n_1 \times n_2 \times n_3}$, the objective function $\frac{1}{2}\|\mathcal{X} - \mathcal{G}\|_F^2 + \xi\|\mathcal{X}\|_{C_{S_{2/3}}^\tau}^{2/3}$ can be reformulated as

$$\begin{aligned}
&\frac{1}{2}\|\mathcal{X} - \mathcal{G}\|_F^2 + \xi\|\mathcal{X}\|_{C_{S_{2/3}}^\tau}^{2/3} \\
=&\frac{1}{2n_3}\|\bar{\mathcal{X}} - \bar{\mathcal{G}}\|_F^2 + \frac{\xi}{n_3}\sum_{j=1}^{n_3}\|\bar{\mathcal{X}}^{(j)}\|_{C_{S_{2/3}}^\tau}^{2/3} \\
=&\frac{1}{n_3}\sum_{j=1}^{n_3}\left(\frac{1}{2}\|\bar{\mathcal{X}}^{(j)} - \bar{\mathcal{G}}^{(j)}\|_F^2 + \xi\|\bar{\mathcal{X}}^{(j)}\|_{C_{S_{2/3}}^\tau}^{2/3}\right).
\end{aligned} \tag{41}$$

Thus, the original tensor optimization problem can be decoupled into $n_3$ independent complex matrix optimization problems for each slice, i.e.,

$$\underset{\bar{\mathcal{X}}^{(j)}}{\arg\min} \frac{1}{2}\|\bar{\mathcal{X}}^{(j)} - \bar{\mathcal{G}}^{(j)}\|_F^2 + \xi\|\bar{\mathcal{X}}^{(j)}\|_{C_{S_{2/3}}^\tau}^{2/3} \tag{42}$$

for $1 \le j \le n_3$.

Denote the SVD of $\bar{\mathcal{G}}^{(j)}$ is $\bar{\mathcal{G}}^{(j)} = \bar{\mathcal{U}}^{(j)}\bar{\mathcal{S}}^{(j)}(\bar{\mathcal{V}}^{(j)})^H$, then, according to Lemma A.1, the optimal solution of (42) is

$$\bar{\mathcal{X}}^{*(j)} = \bar{\mathcal{U}}^{(j)} D_{\xi,\tau}(\bar{\mathcal{S}}^{(j)})(\bar{\mathcal{V}}^{(j)})^H, \tag{43}$$

where $D_{\xi,\tau}(\bar{\mathcal{S}}_{i,i}^{(j)})$ is given by solving the following problem

$$\underset{\sigma_i \ge 0}{\arg\min} \frac{1}{2}\left(\sigma_i - \bar{\mathcal{S}}_{i,i}^{(j)}\right)^2 + \xi \min(\sigma_i, \tau)^{\frac{2}{3}}. \tag{44}$$

$\square$

### A.2. Proofs of Proposition 1

Before proving the Proposition 1, we introduce Lemma A.3 with the goal of restricting the range of regularization parameter $\xi$.

**Lemma A.3.** *For given function*

$$h(\alpha) = \begin{cases} \frac{1}{2}(\alpha - \delta)^2 + \xi\alpha^{\frac{2}{3}}, & \text{if } \alpha \in [0, \tau], \\ \frac{1}{2}(\alpha - \delta)^2 + \xi\tau^{\frac{2}{3}}, & \text{if } \alpha \in (\tau, \infty), \end{cases} \tag{45}$$

$h''(\tau) > 0$ *if and only if* $\xi \in (0, 4.5\tau^{\frac{4}{3}})$.

*Proof.* First, we prove the sufficiency of Lemma A.3. We know that $h''(\tau) > 0$ indicates $1 - 4.5\xi\tau^{-\frac{4}{3}} > 0$. Thus, we can obtain the upper bound of $\xi$, i.e., $\xi < 4.5\tau^{\frac{4}{3}}$. On the other hand, the regularization parameter $\xi$ should be positive. Therefore, we have $\xi \in (0, 4.5\tau^{\frac{4}{3}})$.

Then, we prove the necessity of Lemma A.3. Noted that $\xi < 4.5\tau^{\frac{4}{3}}$, we can obtain $4.5\xi\tau^{-\frac{4}{3}} < 1$. Thus, $h''(\tau) = 1 - 4.5\xi\tau^{-\frac{4}{3}} > 0$. $\square$

Now we are ready to prove Proposition 1.

*Proof.* For given function $h(\alpha)$ in (45), $h'''(\alpha)$ in the range of $(0, \tau]$, $\xi \in (0, 4.5\tau^{\frac{4}{3}})$ indicate $h'''(\alpha) > 0$. Thus, $h''(\alpha)$ is strictly monotonically increasing in the range of $(0, \tau]$. According to Lemma A.3, we know that $h''(\tau) > 0$. Moreover, $\lim_{\alpha \to 0} h''(\alpha) = -\infty$. Thus, there exists a unique point to ensure $h''(\alpha) = 0$ in the range of $(0, \tau)$. According to the fundamental theorem of calculus, $h'(\alpha)$ first decreases and increases subsequently in the range of $(0, \tau]$. Meanwhile, $h'(\alpha)$ strictly increases in the range of $(\tau, \infty)$ because $h''(\alpha) = 1$ when $\alpha > \tau$. Therefore, $h(\alpha)$ can be classified into the following three types (Li et al., 2022a).

**1)** $h(\alpha)$ is non-monotonically increasing in the range of $(0, \tau]$, which corresponding to the case that $h'(\alpha) = 0$ has two different solutions when $\alpha \in (0, \tau]$. We assume that 0 and $x^*$ are the two potential minimal solutions of $h(\alpha)$.

In the case of $h(0) = h(x^*)$, we have $\frac{1}{2}(0 - \delta)^2 + \xi(0)^{\frac{2}{3}} = \frac{1}{2}(x^* - \delta)^2 + \xi(x^*)^{\frac{2}{3}}$ based on (43). From $h'(x^*) = 0$, we can get $\delta = x^* + \frac{2\xi}{3}(x^*)^{-\frac{1}{3}}$. Then, by substituting $\delta$ into the former equation, we can obtain $x^* = \frac{2}{3}\xi^{\frac{3}{4}}$. If let $v = x^*$ and $v' = v + \frac{2\xi}{3}v^{-\frac{1}{3}}$, we can obtain $\delta = v'$ in this case. Next, we consider the minimum value of $h(\alpha)$ in the range of $(\tau, \infty)$, i.e., $h(\delta) = \xi\tau^{\frac{2}{3}}$. By letting $h(\delta) = h(0)$, we can obtain the lower bound of $\tau$, that is, $\tau^* = (\frac{3\delta^2}{4\xi})^2$. Thus, the minimum solution of $h(\alpha)$ in this case is

$$\alpha^* = \begin{cases} \delta, & \text{if } \tau \in [0, \tau^*], \\ \{0, x^*\}, & \text{if } \tau \in (\tau^*, \infty). \end{cases} \tag{46}$$

In the case of $h(0) > h(x^*)$, we let the minimal values of $h(\alpha)$ in the range of $[0, \tau]$ and $(\tau, \infty)$ equal, i.e., $h(x^*) = h(\delta)$. Thus, we have

$$\frac{1}{2}(x^* - \delta)^2 + \xi(x^*)^{\frac{2}{3}} = \xi(\tau)^{\frac{2}{3}}. \tag{47}$$

Through derivation, we can get

$$x^* - \delta + \frac{2\xi}{3}(x^*)^{-\frac{1}{3}} = 0. \tag{48}$$

This equation implies that if and only if $\delta > 0$ it has positive roots and singular values are always non-negative. Hence, we need only to consider $\delta > 0$ from now on. By solving equation (48), Cao et al. (Cao et al., 2013) have showed that $x^* = h_{\xi, \frac{2}{3}}(\delta)$ defined by

$$x' = \begin{cases} \left( \frac{\phi_\xi(\delta) + \Psi_\xi(\delta)}{2} \right)^3, & \text{if } \delta > \frac{2}{3}(24\xi)^{\frac{1}{4}}, \\ 0, & \text{otherwise}, \end{cases} \tag{49}$$

where $\Psi_\xi(\delta) = (2\delta/\phi_\xi(\bar{\mathcal{S}}_{i,i}^{(j)}) - (\phi_\xi(\delta))^2)^{1/2}$ with

$$\phi_\xi(\delta) = \frac{(32\xi)^{\frac{1}{4}}}{\sqrt{3}} \left( \cosh \left( \frac{\operatorname{arccosh} \left( \frac{27}{16}(\delta)^2 (2\xi)^{-\frac{3}{2}} \right)}{3} \right) \right)^{\frac{1}{2}}.$$

Furthermore, the lower bound $\tau^* = \left( \frac{1}{2\xi}(x^* - \delta + (x^*)^{\frac{2}{3}}) \right)^{\frac{3}{2}}$ can be computed by substituting $x^*$ into (47). Thus, the minimum solution of $h(\alpha)$ in this case is

$$\alpha^* = \begin{cases} \delta, & \text{if } \tau \in [0, \tau^*], \\ x^*, & \text{if } \tau \in (\tau^*, \infty). \end{cases} \tag{50}$$

In the case of $h(0) < h(x^*)$, we can get the lower $\tau^* = (\frac{3\delta^2}{4\xi})^2$ by letting $h(0) = h(\delta)$. Thus, the minimum solution of $h(\alpha)$ in this case is

$$\alpha^* = \begin{cases} \delta, & \text{if } \tau \in [0, \tau^*], \\ 0, & \text{if } \tau \in (\tau^*, \infty). \end{cases} \tag{51}$$

**2)** $h(\alpha)$ is a strictly monotonically increasing function in the range of $(0, \tau]$, which corresponding to the case that $h'(\alpha) > 0$ when $\alpha \in (0, \tau]$. Assume that the minimal values of $h(\alpha)$ in the range of $[0, \tau]$ and $(\tau, \infty)$ equal, i.e., $h(0) = h(\alpha)$, we can obtain $\tau^* = (\frac{3\delta^2}{4\xi})^2$. Thus, the minimum solution of $h(\alpha)$ in this case is

$$\alpha^* = \begin{cases} \delta, & \text{if } \tau \in [0, \tau^*], \\ 0, & \text{if } \alpha \in (\tau^*, \infty). \end{cases} \tag{52}$$

**3)** $h(\alpha)$ is a monotonically increasing function in the range of $(0, \tau]$, which corresponding to the case that $h'(\alpha) = 0$ has only one soulution when $\alpha \in (0, \tau]$. The minimum solution of $h(\alpha)$ in this case is similar to the second case.

Finally, we need to prove that $h(0) > h(x^*)$ when $\delta > v'$. By letting $\delta = v' + \epsilon$ where $\epsilon > 0$, we have

$$\begin{aligned} & h(0) - h(x^*) \\ = & \frac{1}{2}(\delta)^2 - \left( \frac{1}{2}(x^* - \delta)^2 + \lambda(x^*)^{\frac{1}{2}} \right) \\ = & \frac{1}{2}(v' + \epsilon)^2 - \left( \frac{1}{2}(x^* - (v' + \epsilon))^2 + \lambda(x^*)^{\frac{1}{2}} \right) \\ = & \epsilon x^* + \left( \frac{1}{2}(v')^2 - \left( \frac{1}{2}(v' - x^*)^2 + \lambda(x^*)^{\frac{1}{2}} \right) \right). \end{aligned} \tag{53}$$

We know that $h(0) = h(x^*)$ when $\delta = v'$, thus, the second term in the last equation of (53) equals 0. We have $h(0) - h(x*) = \epsilon x^*$, i.e., $h(0) > h(x^*)$ when $\delta > v'$.

So, the closed-form global minimum point $\alpha^*$ of $h(\alpha)$ is given by

$$\alpha^* = \begin{cases} \delta, & \text{if } \tau \in [0, \tau^*], \\ h_{\xi, \frac{2}{3}}(\delta), & \text{if } \tau \in (\tau^*, \infty) \end{cases} \tag{54}$$

with $\tau^* = (1/(2\xi)(h_{\xi, \frac{2}{3}}(\delta) - \delta)^2 + (h_{\xi, \frac{2}{3}}(\delta))^{2/3})^{3/2}$. $\qquad\square$

In summary, we can obtain the final solution of $\mathcal{D}_{\xi,\tau}(\bar{\mathcal{S}}_{i,i}^{(j)})$ in (43) when $\xi \in (0, 4.5\tau^{\frac{4}{3}})$, that is

$$\mathcal{D}_{\xi,\tau}(\bar{\mathcal{S}}_{i,i}^{(j)}) = \begin{cases} \bar{\mathcal{S}}_{i,i}^{(j)}, & \text{if } \tau \in [0, \tau^*], \\ x', & \text{if } \tau \in (\tau^*, \infty) \end{cases} \tag{55}$$

with $\tau^* = \left( \frac{1}{2\xi}(x' - \bar{\mathcal{S}}_{i,i}^{(j)})^2 + (x')^{\frac{2}{3}} \right)^{\frac{3}{2}}$ and

$$x' = \begin{cases} \left( \frac{\phi_\xi(\bar{\mathcal{S}}_{i,i}^{(j)}) + \Psi_\xi(\bar{\mathcal{S}}_{i,i}^{(j)})}{2} \right)^3, & \text{if } \bar{\mathcal{S}}_{i,i}^{(j)} > \frac{2}{3}(24\xi)^{\frac{1}{4}}, \\ 0, & \text{otherwise,} \end{cases} \tag{56}$$

where $\Psi_\xi(\bar{\mathcal{S}}_{i,i}^{(j)}) = (2\bar{\mathcal{S}}_{i,i}^{(j)}/\phi_\xi(\bar{\mathcal{S}}_{i,i}^{(j)}) - (\phi_\xi(\bar{\mathcal{S}}_{i,i}^{(j)}))^2)^{1/2}$ with

$$\phi_\xi(\bar{\mathcal{S}}_{i,i}^{(j)}) = \frac{(32\xi)^{\frac{1}{4}}}{\sqrt{3}} \left( \cosh \left( \frac{\text{arccosh} \left( \frac{27}{16}(\bar{\mathcal{S}}_{i,i}^{(j)})^2 (2\xi)^{-\frac{3}{2}} \right)}{3} \right) \right)^{\frac{1}{2}}.$$

.

Finally, once optimal solution $\bar{\mathcal{X}}^*$ is obtained by rule (43), we can perform the inverse FFT operator on $\bar{\mathcal{X}}^*$ to obtain $\mathcal{X}^*$, i.e., $\mathcal{X}^* = \text{ifft}(\bar{\mathcal{X}}^*, [], 3) = \mathcal{U} * \text{ifft}(D_{\xi,\tau}(\bar{\mathcal{S}}), [], 3)\mathcal{V}^T$.

### A.3. The Proof of Convergence Theorem

Before proving Theorem 2, the definition 1 is firstly presented. According to (Chan et al., 2017), we propose that the function $\Omega$ in the plug-and-play module is a bounded denoiser.

**Definition A.1** (Bounded Denoiser (Chan et al., 2017) ). The function $\mathcal{F}_\sigma : \mathbb{R}^n \to \mathbb{R}^n$, defined by the parameter $\sigma$, is a bounded denoiser. For any input $x \in \mathbb{R}^n$, the denoising satisfies the following condition

$$\frac{\|\mathcal{F}_\sigma(x, \sigma) - x\|^2}{n} \leq \sigma^2 C, \tag{57}$$

where $C$ is a universal constant, which is independent of $n$ and $\sigma$.

We assume that the function $\Omega$ in the $\mathcal{Y}$-subproblem is a bounded bounded denoiser. Then, for any input $\mathcal{X} \in \mathbb{R}^{n_1 \times n_2 \times n_3}$, $\|\Omega(\mathcal{X}, \sigma) - \mathcal{X}\|_F$ is bounded. Specially, $\Omega$ is the identity operator when $\sigma \to 0$.

Then, we introduce the following Lemmas.

**Lemma A.4.** *(Lin et al., 2010) Suppose $\mathcal{H}$ is a real Hilbert space with an inner product $\langle \cdot, \cdot \rangle$, the norm $\| \cdot \|$ has its dual norm $\| \cdot \|^{dual}$, and $y \in \partial \|x\|$, where $\partial f(\cdot)$ is the subgradient of function $f(\cdot)$. Then $\|y\|^{dual} = 1$ if $x \neq 0$, while $\|y\|^{dual} \leq 1$ if $x = 0$.*

**Lemma A.5.** *(Lewis & Sendov, 2005) Let $F(\cdot) : \mathbb{R}^{m \times n} \to \mathbb{R}$ be represented as $F(X) = f \circ \sigma(X) = f(\sigma_1(X), \sigma_2(X), .., \sigma_r(X))$, where the function $f(\cdot) : \mathbb{R}^n \to \mathbb{R}$ is differentiable, $X = U diag(\sigma_1, \sigma_2, ..., \sigma_r)V^T$ is SVD of $X \in \mathbb{R}^{m \times n}$, and $r = min(m, n)$. Then the gradient of $F(X)$ at matrix $X$ is*

$$\frac{\partial F(X)}{\partial X} = \partial(f \circ \sigma)(X) = U diag(\Delta)V^T, \tag{58}$$

where $\Delta = \partial f(\sigma(X)) = (\frac{\partial f(\sigma_1(X))}{\partial X}, ..., \frac{\partial f(\sigma_r(X))}{\partial X})$ and $\frac{\partial f(\sigma_i(X))}{\partial X} = \partial|\sigma_i(X)|f'[\sigma_i(X)]$, $i = 1, ..., r$.

**Lemma A.6** (Subdifferential of capped tensor Schatten 2/3 norm)**.** *For the proposed capped tensor Schatten 2/3 norm* $\|\mathcal{G}\|_{C^{\tau}_{S_{2/3}}}^{2/3}$ *with non-convex penalty function* $f(x) = min(x, \tau)^{\frac{2}{3}}$*, its subdifferential at* $\mathcal{X}$ *is*

$$\partial\|\mathcal{G}\|_{C^{\tau}_{S_{2/3}}}^{2/3} = \frac{1}{n}\mathcal{U} * \text{ifft}(\partial f(\bar{\mathcal{S}}), [], 3) * \mathcal{V}^T, \tag{59}$$

*where* $\partial f(\bar{\mathcal{S}})$ *is a f-diagonal tensor, in which* $i$*-th diagonal element is* $\partial f(\bar{\mathcal{S}}_{i,i}^{(j)}) = c(j, i)f'(\bar{\mathcal{S}}_{i,i}^{(j)})$*, where*

$$c(j, i) = \begin{cases} 1, & if\ \bar{\mathcal{S}}_{i,i}^{(j)} > 0, \\ [-1, 1], & if\ \bar{\mathcal{S}}_{i,i}^{(j)} = 0. \end{cases} \tag{60}$$

In practice, $f(x) = min(x, \tau)^{\frac{2}{3}}$ is approximated by $f(x) = min(x + \epsilon, \tau)^{\frac{2}{3}}$, where $\epsilon \to 0$ is a small positive constant to avoid division by zero. Hence, we can obtain

$$\partial f(\bar{\mathcal{S}}_{i,i}^{(j)}) = \begin{cases} \frac{2}{3}(\bar{\mathcal{S}}_{i,i}^{(j)} + \epsilon)^{-\frac{1}{3}}, & if\ \bar{\mathcal{S}}_{i,i}^{(j)} \leq \tau, \\ 0, & otherwise\ . \end{cases} \tag{61}$$

**Lemma A.7.** *(Marjanovic & Solo, 2012) For* $\bar{\mathcal{S}}_{i,i}^{(j)} > \frac{2}{3}(24\xi)^{\frac{1}{4}}$ *in (56), we have* $\left((\phi_\xi(\bar{\mathcal{S}}_{i,i}^{(j)}) + \Psi_\xi(\bar{\mathcal{S}}_{i,i}^{(j)}))/2\right)^3 \to \bar{\mathcal{S}}_{i,i}^{(j)}$ *from blew as* $\bar{\mathcal{S}}_{i,i}^{(j)} \to \infty$.

After introducing these lemmas, we can prove Theorem 2.

*Proof.* 1) Firstly, we will prove the boundedness of sequence $\{\Gamma_k\}_{k=1}^{\infty}$. For $\{\mathcal{Y}_{3,k}\}$, we have $\mathcal{Y}_{k+1} = \Omega(\mathcal{Z}_{k+1} + \frac{\mathcal{Y}_{3,k}}{\mu_{3,k}}, \frac{\lambda_2}{\mu_{3,k}})$. Based on our assumption, we can deduce

$$\left\|\mathcal{Y}_{k+1} - \left(\mathcal{Z}_{k+1} + \frac{\mathcal{Y}_{3,k}}{\mu_{3,k}}\right)\right\|_F \leq \frac{\lambda_2}{\mu_{3,k}}C', \tag{62}$$

where $C'$ is a constant, which is independent of $\mu_3$ and $\lambda_2$, combining with the updating rule of $\mathcal{Y}_k$, we have

$$\|\mathcal{Y}_{3,k+1}\|_F = \|\mathcal{Y}_{3,k} + \mu_{3,k}(\mathcal{Z}_{k+1} - \mathcal{Y}_{k+1})\|_F \leq \lambda_2 C'. \tag{63}$$

Hence, $\{\mathcal{Y}_{3,k}\}$ is bounded.

According to the first-order optimal condition of $\mathcal{G}$ and the updating rule of $\mathcal{Y}_{2,k}$, we can obtain

$$\begin{aligned} \mathbf{0} &\in \partial\|\mathcal{G}\|_{C^{\tau}_{S_{2/3}}}^{2/3} - \mathcal{Y}_{2,k} - \mu_{2,k}(\mathcal{Z}_{k+1} - \mathcal{G}_{k+1}) \\ &= \partial\|\mathcal{G}\|_{C^{\tau}_{S_{2/3}}}^{2/3} - \mathcal{Y}_{2,k+1}. \end{aligned} \tag{64}$$

Assume that the t-SVD of tensor $\mathcal{G} \in \mathbb{R}^{n \times V \times n}$ is $\mathcal{U} * \mathcal{S} * \mathcal{V}^T$ and $\bar{\mathcal{G}} = \text{fft}(\mathcal{G}, [], 3)$. By using Lemma A.5, we can deduce that

$$\begin{aligned} \|\partial\|\mathcal{G}\|_{C^{\tau}_{S_{2/3}}}^{2/3}\|_F^2 &= \|\frac{1}{n}\mathcal{U} * \text{ifft}(\partial f(\bar{\mathcal{S}}), [], 3) * \mathcal{V}^T\|_F^2 \\ &= \frac{1}{n^2}\|\text{ifft}(\partial f(\bar{\mathcal{S}}), [], 3)\|_F^2 \\ &= \frac{1}{n^2}\frac{1}{n}\|\partial f(\bar{\mathcal{S}})\|_F^2 \\ &= \frac{1}{n^3}\sum_{j=1}^{n}\sum_{i=1}^{V}[\partial f(\bar{\mathcal{S}}_{i,i}^{(j)})]^2 \\ &\leq \frac{1}{n^3} \cdot n \cdot V \cdot (\frac{2}{3}\epsilon^{-\frac{1}{3}})^2 = \frac{4V}{9n^2\epsilon^{\frac{2}{3}}}, \end{aligned} \tag{65}$$

where $V \ll n$. From the above derivation, we can see that $\partial \|\mathcal{G}\|_{C^\tau_{S_{2/3}}}^{2/3}$ is bounded. Therefore, from (64), it is easy to verify boundedness of $\{\mathcal{Y}_{2,k}\}$.

According to the updating rule of $E$ in Algorithm 1 and the first-order optimal condition, we can obtain

$$0 \in \frac{\lambda_1}{\mu_{1,k}} \partial \|E_{k+1}\|_{C^\theta_{2,2/3}}^{2/3} + E_{k+1} - D_{k+1}, \tag{66}$$

where $D_{k+1} = [X^{(1)} - X^{(1)}Z_{k+1}^{(1)} + Y_{1,k}^{(1)}/\mu_{1,k}; ...; X^{(V)} - X^{(V)}Z_{k+1}^{(V)} + Y_{1,k}^{(V)}/\mu_{1,k}]$. Notice that the updating rule of $Y_1$ implies $Y_{1,k+1} = \mu_{1,k}(D_{k+1} - E_{k+1})$, we can deduce

$$0 \in \lambda_1 \partial \|E_{k+1}\|_{C^\theta_{2,2/3}}^{2/3} - Y_{k+1}, \tag{67}$$

where $Y_{k+1} = [Y_{k+1}^{(1)}; ...; Y_{k+1}^{(V)}]$. Thus, we have

$$Y_{1,k+1}(:,j) \in \lambda_1 \partial \|E_{k+1}\|_{C^\theta_{2,2/3}}^{2/3}(:,j), \tag{68}$$

where $Y_{k+1}(:,j)$ and $E_{k+1}(:,j)$ are the $j$-th column of $Y_{1,k+1}$ and $E_{k+1}$, respectively. Then, the $j$-th column of $\partial \|E_{k+1}\|_{C^\theta_{2,2/3}}^{2/3}$ is

$$\begin{cases} \frac{2E_{k+1}(:,j)}{3\|E_{k+1}(:,j)\|_2^{4/3}}, & \text{if } 0 < \|E_{k+1}(:,j)\|_2^{2/3} < \theta \\ 0, & \text{otherwise} . \end{cases} \tag{69}$$

Thus, $\partial \|E_{k+1}\|_{C^\theta_{2,2/3}}^{2/3}$ is bounded. Note that $l_2$ norm is self-dual, according to Lemma A.5 and (68), we can obtain that $\|Y_{1,k+1}(:,j)\|_2 \leq \lambda_1$, i.e., $\{Y_{1,k}^{(v)}\}$ is bounded.

By the iterative scheme in Algorithm 1, we can deduce

$$
\begin{aligned}
&\mathcal{L}_{\mu_{i,k}}(\{Z_{k+1}^{(v)}\}_{v=1}^V, E_{k+1}, \mathcal{G}_{k+1}, \mathcal{Y}_{k+1}; \{Y_{1,k}^{(v)}\}_{v=1}^V, \mathcal{Y}_{2,k}, \mathcal{Y}_{3,k}) \\
&\leq \mathcal{L}_{\mu_{i,k}}(\{Z_k^{(v)}\}_{v=1}^V, E_k, \mathcal{G}_k, \mathcal{Y}_k; \{Y_{1,k}^{(v)}\}_{v=1}^V, \mathcal{Y}_{2,k}, \mathcal{Y}_{3,k}) \\
&= \mathcal{L}_{\mu_{i,k-1}}(\{Z_k^{(v)}\}_{v=1}^V, E_k, \mathcal{G}_k, \mathcal{Y}_k; \{Y_{1,k-1}^{(v)}\}_{v=1}^V, \mathcal{Y}_{2,k-1}, \mathcal{Y}_{3,k-1}) + \frac{\mu_{2,k} - \mu_{2,k-1}}{2}\|\mathcal{Z}_k - \mathcal{G}_k\|_F^2 \\
&\quad + \sum_{v=1}^V \left( \frac{\mu_{1,k} - \mu_{1,k-1}}{2}\|X^{(v)} - X^{(v)}Z_k^{(v)} - E_k^{(v)}\|_F^2 + \left\langle Y_{1,k}^{(v)} - Y_{1,k-1}^{(v)}, X^{(v)} - X^{(v)}Z_k^{(v)} - E_k^{(v)}\right\rangle \right) \\
&\quad + \langle \mathcal{Y}_{2,k} - \mathcal{Y}_{2,k-1}, \mathcal{Z}_k - \mathcal{G}_k\rangle + \langle \mathcal{Y}_{3,k} - \mathcal{Y}_{3,k-1}, \mathcal{Z}_k - \mathcal{Y}_k\rangle + \frac{\mu_{3,k} - \mu_{3,k-1}}{2}\|\mathcal{Z}_k - \mathcal{Y}_k\|_F^2 \\
&= \mathcal{L}_{\mu_{i,k-1}}(\{Z_k^{(v)}\}_{v=1}^V, E_k, \mathcal{G}_k, \mathcal{Y}_k; \{Y_{1,k-1}^{(v)}\}_{v=1}^V, \mathcal{Y}_{2,k-1}, \mathcal{Y}_{3,k-1}) + \frac{\mu_{2,k} + \mu_{2,k-1}}{2\mu_{2,k-1}^2}\|\mathcal{Y}_{2,k} - \mathcal{Y}_{2,k-1}\|_F^2 \\
&\quad + \sum_{v=1}^V \frac{\mu_{1,k} + \mu_{1,k-1}}{2\mu_{1,k-1}^2}\|Y_{1,k}^{(v)} - Y_{1,k-1}^{(v)}\|_F^2 + \frac{\mu_{3,k} + \mu_{3,k-1}}{2\mu_{3,k-1}^2}\|\mathcal{Y}_{3,k} - \mathcal{Y}_{3,k-1}\|_F^2
\end{aligned}
\tag{70}
$$

with $i = 1, 2, 3$ and

$$
\begin{aligned}
&\mathcal{L}_{\mu_{i,k}}(\{Z_{k+1}^{(v)}\}_{v=1}^V, E_{k+1}, \mathcal{G}_{k+1}, \mathcal{Y}_{k+1}; \{Y_{1,k}^{(v)}\}_{v=1}^V, \mathcal{Y}_{2,k}, \mathcal{Y}_{3,k}) \\
&\quad + \frac{1}{2\mu_{1,k}}\|Y_{1,k}^{(v)}\|_F^2 + \frac{1}{2\mu_{2,k}}\|\mathcal{Y}_{2,k}\|_F^2 + \frac{1}{2\mu_{3,k}}\|\mathcal{Y}_{3,k}\|_F^2 \\
&= \|\mathcal{G}_{k+1}\|_{C_{S_{2/3}}^\tau}^{2/3} + \lambda_1\|E_{k+1}\|_{C_{2,2/3}^\theta}^{2/3} + \lambda_2\Theta(\mathcal{Y}_{k+1}) \\
&\quad + \sum_{v=1}^V \frac{\mu_{1,k}}{2}\left\|X^{(v)} - X^{(v)}Z_{k+1}^{(v)} - E_k^{(v)} + \frac{Y_{1,k}^{(v)}}{\mu_{1,k}}\right\|_F^2 \\
&\quad + \frac{\mu_{2,k}}{2}\|\mathcal{Z}_{k+1} - \mathcal{G}_{k+1} + \frac{\mathcal{Y}_{2,k}}{\mu_k}\|_F^2 \\
&\quad + \frac{\mu_{3,k}}{2}\|\mathcal{Z}_{k+1} - \mathcal{Y}_{k+1} + \frac{\mathcal{Y}_{3,k}}{\mu_k}\|_F^2.
\end{aligned}
\tag{71}
$$

Summing result (70) from $k = 1$ to $n(\geq 1)$, we have

$$
\begin{aligned}
&\mathcal{L}_{\mu_{i,k}}(\{Z_{k+1}^{(v)}\}_{v=1}^V, E_{k+1}, \mathcal{G}_{k+1}, \mathcal{Y}_{k+1}; \{Y_{1,k}^{(v)}\}_{v=1}^V, \mathcal{Y}_{2,k}, \mathcal{Y}_{3,k}) \\
&\leq \mathcal{L}_{\mu_{i,0}}(\{Z_1^{(v)}\}_{v=1}^V, E_1, \mathcal{G}_1, \mathcal{Y}_1; \{Y_{1,0}^{(v)}\}_{v=1}^V, \mathcal{Y}_{2,0}, \mathcal{Y}_{3,0}) \\
&\quad + \sum_{k=1}^n \frac{\mu_{1,k} + \mu_{1,k-1}}{2\mu_{1,k-1}^2}\sum_{v=1}^V\left\|Y_{1,k}^{(v)} - Y_{1,k-1}^{(v)}\right\|_F^2 \\
&\quad + \sum_{k=1}^n \frac{\mu_{2,k} + \mu_{2,k-1}}{2\mu_{2,k-1}^2}\|\mathcal{Y}_{2,k} - \mathcal{Y}_{2,k-1}\|_F^2 \\
&\quad + \sum_{k=1}^n \frac{\mu_{3,k} + \mu_{3,k-1}}{2\mu_{3,k-1}^2}\|\mathcal{Y}_{3,k} - \mathcal{Y}_{3,k-1}\|_F^2.
\end{aligned}
\tag{72}
$$

Here, we can drive that

$$
\sum_{k=1}^n \frac{\mu_{i,k} + \mu_{i,k-1}}{2\mu_{i,k-1}^2} \leq \frac{\eta(\eta+1)}{2\mu_{i,0}(\eta-1)} < +\infty, \quad i = 1, 2, 3
\tag{73}
$$

where (73) holds because of $\mu_{i,k} = \eta\mu_{i,k-1}$. At the same time, taking the boundedness of $\{Y_{1,k}^{(v)}\}$, $\{\mathcal{Y}_{2,k}\}$ and $\{\mathcal{Y}_{3,k}\}$ into consideration, we can naturally conclude the boundedness of $\mathcal{L}_{\mu_{i,k}}(\{Z_{k+1}^{(v)}\}_{v=1}^V, E_{k+1}, \mathcal{G}_{k+1}, \mathcal{Y}_{k+1}; \{Y_{1,k}^{(v)}\}_{v=1}^V,$ $\mathcal{Y}_{2,k}, \mathcal{Y}_{3,k})$. Then, the left side of (71) is bounded. Further, we can deduce that right side of (71) is bounded, that is each term of (71) is bounded and nonnegative, i.e., $\{\mathcal{G}_k\}$, $\{E_k\}$, $\{Z_k^{(v)}\}$ and $\{\mathcal{Y}_k\}$ are bounded. In summary, we can conclude that The sequence $\{\mathcal{T}_k\}_{k=1}^\infty$ is bounded.

2) Based on the boundedness of sequence $\{\mathcal{T}_k\}_{k=1}^\infty$ and Bolzano-Weierstrass theorem (Bartle & Sherbert, 2000) that every bounded sequence must have a convergent subsequence.

For $\{\mathcal{G}_k\}$, given the boundedness of the sequences $\{\mathcal{Y}_{2,k}\}$, with $\lim_{k\to\infty}\mu_k = \infty$, we have

$$
\begin{aligned}
&\|\mathcal{G}_{k+1} - \mathcal{G}_k\|_F \\
&= \left\|\mathcal{N}_{\frac{1}{\mu_{2,k}},\tau}(\mathcal{A}_k) - \mathcal{G}_k\right\|_F \\
&= \left\|\mathcal{N}_{\frac{1}{\mu_{2,k}},\tau}(\mathcal{A}_k) - \mathcal{A}_k + \frac{\mathcal{Y}_{2,k}}{\mu_{2,k}} + \frac{\mathcal{Y}_{2,k} - \mathcal{Y}_{2,k-1}}{\mu_{2,k-1}}\right\|_F \\
&\leq \left\|\mathcal{N}_{\frac{1}{\mu_{2,k}},\tau}(\mathcal{A}_k) - \mathcal{A}_k\right\|_F + \left\|\frac{\mathcal{Y}_{2,k}}{\mu_{2,k}} + \frac{\mathcal{Y}_{2,k} - \mathcal{Y}_{2,k-1}}{\mu_{2,k-1}}\right\|_F,
\end{aligned}
\tag{74}
$$

where $\mathcal{N}_{\frac{1}{\mu_{2,k}},\tau}(\mathcal{A}_k)$ represents the solution to the subproblem $\mathcal{G}$ at $(k+1)$-th iteration. By Lemma A.1, Lemma A.7 and Theorem 1, we know that

(a) If $\tau \in [0, \tau^*]$, $\mathcal{D}_{\xi,\tau}(\bar{\mathcal{S}}_{i,i}^{(j)}) = \bar{\mathcal{S}}_{i,i}^{(j)}$.

(b) If $\tau \in [\tau^*, \infty)$ and $\bar{\mathcal{S}}_{i,i}^{(j)} \in (0, \frac{2}{3}(24\xi)^{\frac{1}{4}})$, $\mathcal{D}_{\xi,\tau}(\bar{\mathcal{S}}_{i,i}^{(j)}) = 0$.

(c) If $\tau \in [\tau^*, \infty)$ and $\bar{\mathcal{S}}_{i,i}^{(j)} \in (\frac{2}{3}(24\xi)^{\frac{1}{4}}, \infty)$, $\mathcal{D}_{\xi,\tau}(\bar{\mathcal{S}}_{i,i}^{(j)}) = \left( \frac{\phi_\xi(\bar{\mathcal{S}}_{i,i}^{(j)}) + \Psi_\xi(\bar{\mathcal{S}}_{i,i}^{(j)})}{2} \right)^3$.

We define $\Delta = |\mathcal{D}_{\xi,\tau}(\bar{\mathcal{S}}_{i,i}^{(j)}) - \bar{\mathcal{S}}_{i,i}^{(j)}|$. By the above discussions, we know that

- $\Delta = 0$ in case (a);

- $\Delta = |\bar{\mathcal{S}}_{i,i}^{(j)}| < \frac{2}{3}(24\xi)^{\frac{1}{4}}$ in case (b);

- According to Lemma A.3, $\Delta = \left| \mathcal{D}_{\xi,\tau}(\bar{\mathcal{S}}_{i,i}^{(j)}) - \bar{\mathcal{S}}_{i,i}^{(j)} \right| \to 0$ as $\bar{\mathcal{S}}_{i,i}^{(j)} \to \infty$ for any $j$ in case (c). We define

$$F(\sigma_i) = \frac{1}{2}(\sigma_i - \bar{S}_{i,i}^{(j)})^2 + \xi \min(\sigma_i, \tau)^{\frac{2}{3}}. \tag{75}$$

Assume that $\sigma^* = \mathcal{D}_{\xi,\tau}(\bar{\mathcal{S}}_{i,i}^{(j)})$, then $F(\sigma^*) \le F(\bar{S}_{i,i}^{(j)})$, we have

$$\frac{1}{2}(\sigma^* - \bar{S}_{i,i}^{(j)})^2 + \xi \min(\sigma^*, \tau)^{\frac{2}{3}} \le \xi \min(\bar{S}_{i,i}^{(j)}, \tau)^{\frac{2}{3}}. \tag{76}$$

Remove non-negative terms, we can deduce

$$\frac{1}{2}(\sigma^* - \bar{S}_{i,i}^{(j)})^2 \le \xi \min(\bar{S}_{i,i}^{(j)}, \tau)^{\frac{2}{3}}. \tag{77}$$

Then,

$$|\sigma^* - \bar{S}_{i,i}| \le \sqrt{2\xi}(\min(\bar{S}_{i,i}^{(j)}, \tau))^{\frac{1}{3}} \le \sqrt{2\xi}(\bar{S}_{i,i}^{(j)})^{\frac{1}{3}}. \tag{78}$$

In summary, we have $\Delta \le max\{\frac{2}{3}(24\xi)^{\frac{1}{4}}, \sqrt{2\xi}(\bar{S}_{i,i}^{(j)})^{\frac{1}{3}}\}$
$= C$, if $\bar{S}_{i,i}^{(j)}$ is bounded. Otherwise, $\Delta \le \frac{2}{3}(24\xi)^{\frac{1}{4}} = C$.

Let the t-SVD of $\mathcal{A}_k$ be $\mathcal{U}_{A_k} * \mathcal{S}_{A_k} * \mathcal{V}_{A_k}^T$ and $\mu_k^{-1} = \xi$, we can deduce that

$$\begin{aligned}
&\|\mathcal{G}_{k+1} - \mathcal{G}_k\|_F \\
&\le \frac{1}{N} \left\| \mathcal{D}_{\frac{1}{\mu_k},\tau}(\bar{\mathcal{S}}_{A_k}) - \bar{\mathcal{S}}_{A_k} \right\|_F + \left\| \frac{\mathcal{Y}_{2,k}}{\mu_{2,k}} + \frac{\mathcal{Y}_{2,k} - \mathcal{Y}_{2,k-1}}{\mu_{2,k-1}} \right\|_F \\
&= \frac{1}{N} \sum_{v=1}^{N} \left\| \mathcal{D}_{\frac{1}{\mu_k},\tau}(\bar{\mathcal{S}}_{A_k}^{(j)}) - \bar{\mathcal{S}}_{A_k}^{(j)} \right\|_F + \left\| \frac{\mathcal{Y}_{2,k}}{\mu_{2,k}} + \frac{\mathcal{Y}_{2,k} - \mathcal{Y}_{2,k-1}}{\mu_{2,k-1}} \right\|_F \\
&\le C\sqrt{V} + \left\| \frac{\mathcal{Y}_{2,k}}{\mu_{2,k}} + \frac{\mathcal{Y}_{2,k} - \mathcal{Y}_{2,k-1}}{\mu_{2,k-1}} \right\|_F,
\end{aligned} \tag{79}$$

where the second inequality is the fact $V \ll N$. Therefore, we can obtain $\lim_{k\to\infty} \|\mathcal{G}_{k+1} - \mathcal{G}_k\|_F = 0$ when $\lim_{k\to\infty} \mu_k = \infty$, that is $\{\mathcal{G}_k\}$ converges.

For $\{Z_k^{(v)}\}$, we can draw the following conclusion

$$\lim_{k\to\infty} (\mathcal{Z}_{k+1} - \mathcal{G}_{k+1}) = \lim_{k\to\infty} \frac{\mathcal{Y}_{2,k+1} - \mathcal{Y}_{2,k}}{\mu_{2,k}} = 0. \tag{80}$$

Then, we have

$$\begin{aligned}
&\|\mathcal{Z}_{k+1} - \mathcal{Z}_k\|_F \\
&= \|\mathcal{Z}_{k+1} - \mathcal{G}_{k+1} + \mathcal{G}_{k+1} - \mathcal{G}_k + \mathcal{G}_k - \mathcal{Z}_k\|_F \\
&\le \|\mathcal{Z}_{k+1} - \mathcal{G}_{k+1}\|_F + \|\mathcal{G}_{k+1} - \mathcal{G}_k\|_F + \|\mathcal{G}_k - \mathcal{Z}_k\|_F.
\end{aligned} \tag{81}$$

*Table 9.* Comparison with deep multi-view clustering methods on BBC4View, Notting-Hill, and UCI-digits datasets.

| Method | BBC4View | | | Notting-Hill | | | UCI-digits | | |
|---|---|---|---|---|---|---|---|---|---|
| | ACC | NMI | AR | ACC | NMI | AR | ACC | NMI | AR |
| SCMVC | 0.9226 | 0.8661 | 0.8454 | 0.8328 | 0.8031 | 0.7800 | 0.9345 | 0.8751 | 0.8617 |
| SSLNMVC | 0.9058 | 0.8953 | 0.8221 | 0.8187 | 0.7828 | 0.7597 | 0.9058 | 0.8953 | 0.8556 |
| Ours | **0.9985** | **0.9945** | **0.9971** | **0.9983** | **0.9924** | **0.9959** | **1.0000** | **1.0000** | **1.0000** |

Thus, we can get $\lim_{k\to 0} \|\mathcal{Z}_{k+1} - \mathcal{Z}_k\|_F$, that is $\{Z_k^{(v)}\}$ converges.

For $\{\mathcal{Y}_k\}$, we have

$$\lim_{k\to\infty} (\mathcal{Z}_{k+1} - \mathcal{Y}_{k+1}) = \lim_{k\to\infty} \frac{\mathcal{Y}_{3,k+1} - \mathcal{Y}_{3,k}}{\mu_{3,k}} = 0. \tag{82}$$

Similarly, we get $\lim_{k\to 0} \|\mathcal{Y}_{k+1} - \mathcal{Y}_k\|_F = 0$.

For $\{E_k^{(v)}\}$, we can deduce

$$\lim_{k\to\infty} (X^{(v)} - X^{(v)} Z_{k+1}^{(v)} - E_{k+1}^{(v)}) = \lim_{k\to\infty} \frac{Y_{1,k+1}^{(v)} - Y_{1,k}^{(v)}}{\mu_{1,k}} = 0. \tag{83}$$

Thus, we also have $\lim_{k\to 0} \left\| E_{k+1}^{(v)} - E_k^{(v)} \right\| = 0$.

Finally, sequences $\{\mathcal{G}_k\}$, $\{Z_k^{(v)}\}$, $\{\mathcal{Y}_k\}$ and $\{E_k^{(v)}\}$ generated by Algorithm 1 converge, and proofs complete. □

## B. More Experiment Results

In this section, we present the experimental results of other comparison methods.

**Comparison with deep multiview methods.** To demonstrate the effectiveness of the proposed method in comparison with representative deep multi-view clustering approaches, the clustering results obtained by our method, SCMVC(Luo et al., 2025), and SSLNMVC(Yan et al., 2025) on the BBC4View, Notting-Hill, and UCI-digits datasets are reported in Table 9. It can be observed that our method achieves competitive or superior results without requiring heavy training overhead. Specifically, our method demonstrates a substantial margin of improvement across all three evaluation metrics. For instance, on the Notting-Hill dataset, our method exceeds the second-best performing model by over 16.55% in terms of ACC. These results convincingly verify that our approach can capture the underlying data structure more effectively than these deep learning-based methods, indicating that the integration of accurate nonconvex low-rank modeling and deep priors can recover more discriminative and structurally coherent representations without requiring costly network training.

**Visualization of Embedded Representation.** The embedding results generated by our method and four comparison methods on Yale and COIL20 datasets are exhibited in Fig. 7. It can be observed that the samples in Fig. 7(a) - Fig. 7(d) are scattered and disordered, while our method yields clearly separable clusters. This also clearly validates that our proposed method can recover coherent patterns with a compact and separable cluster structure.

**Confusion Matrices.** The confusion matrices for clustering results obtained by three methods on the BBC4View and Yale datasets are visualized. We can see that the confusion matrices in Fig. 8(a) and Fig. 8(c) exhibit noticeable misclassification in some categories, while our method achieves perfect diagonal dominance, indicating that there is no confusion among different clusters. This advantage is attributed to the ability of our learned representations to encode the cluster structure effectively.

**Affinity Matrices.** The affinity graphs constructed by four methods on the UCI-Digits and COIL20 datasets are visualized in Fig. 9. It can be seen that the block diagonal structures for affinity matrices in Fig. 9(a) - Fig. 9(c) are obviously indistinct, while our method yields a more apparent block diagonal structure. This indicates that our method can reveal more discriminative subspace information and intrinsic clustering structure of data well.

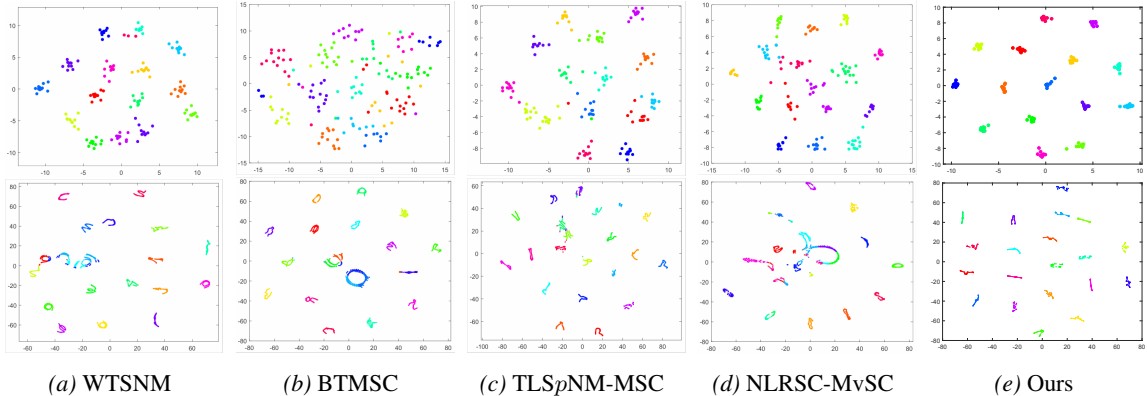

*(a)* WTSNM     *(b)* BTMSC     *(c)* TLS*p*NM-MSC     *(d)* NLRSC-M*v*SC     *(e)* Ours

*Figure 7.* Visualization results of different methods on Yale dataset (the first row) and COIL20 dataset (the second row).

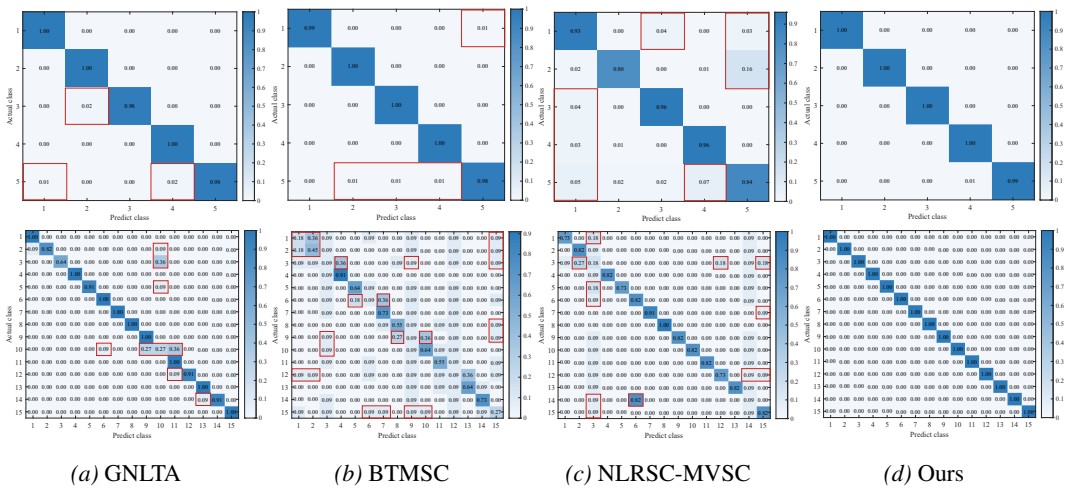

*(a)* GNLTA     *(b)* BTMSC     *(c)* NLRSC-MVSC     *(d)* Ours

*Figure 8.* The confusion matrices comparison among six methods on BBC4View dataset (the first row) and Yale dataset (the second row).

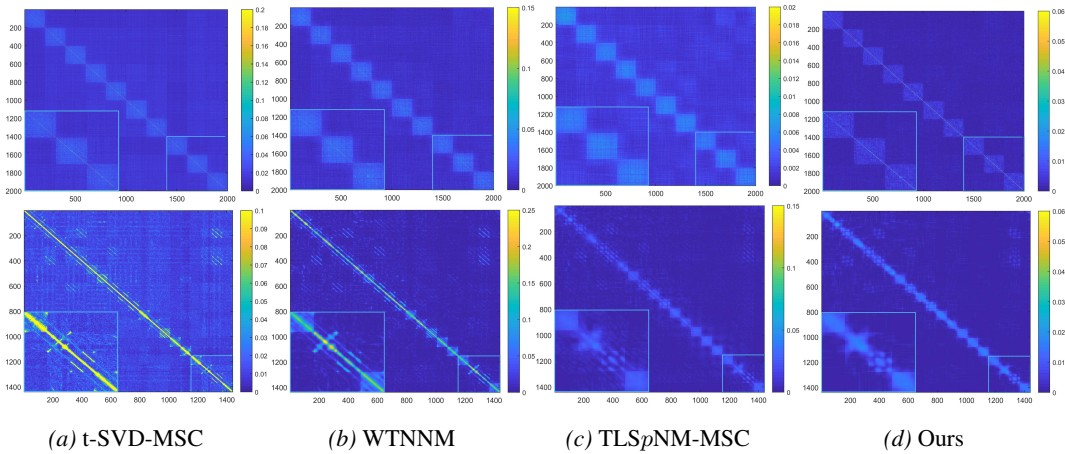

*(a)* t-SVD-MSC     *(b)* WTNNM     *(c)* TLS*p*NM-MSC     *(d)* Ours

*Figure 9.* Comparison of affinity matrices of different methods on UCI-digits dataset (the first row) and COIL20 dataset (the second row).

