# OpenReview forum: "Nonconvex Low-Rank Tensor Representation with Deep Priors for Multiview Subspace Clustering"
_ICML.cc/2026/Conference — ICML 2026 regular_

### Official Review · Reviewer_toG1 · 2026-03-02

**Soundness:** 3
**Presentation:** 3
**Significance:** 3
**Originality:** 3
**Overall Recommendation:** 5
**Confidence:** 5

**Summary:**

This paper proposed a tensorial multi-view clustering method, which proposed a new approximation function for the low-rank tensor learning framework, termed as capped tensor Schatten 2/3-norm. Extensive experiments are  conducted to prove the effectiveness.

**Compliance With Llm Reviewing Policy:**

Affirmed.

**Final Justification:**

I would like to sincerely thank the authors for their extensive efforts and patience during the rebuttal and discussion phases. In their second reply, the authors directly and effectively addressed my core concerns regarding the practical advantages of their proposed non-convex function and the validity of the experimental settings.

My initial core concerns have been successfully resolved based on the following points:

- Superiority of the Non-Convex Function Validated: The authors provided a fair, apples-to-apples comparison by replacing the low-rank term within the same t-SVD-MSC baseline framework. The newly provided experimental results convincingly demonstrate that the proposed regularizer holds a distinct advantage in robustness when handling noisy data compared to existing non-convex functions (such as WTNN, WTSN, and TLSpN). I fully acknowledge and validate this advantage.

- Robustness on Shuffled Datasets: My primary goal in raising the issue of data ordering was to remind the community that tensor-based multi-view clustering methods must be validated on randomly permuted datasets to reflect real-world, unstructured scenarios. I highly appreciate that the authors receptively adopted my feedback, utilized the provided shuffling code, and re-evaluated their method. The new results demonstrate that while the absence of the pre-sorted prior understandably causes a performance drop across tensor methods, the proposed framework still maintains a competitive edge and outperforms state-of-the-art unstructured methods. I believe this mathematically robust framework, along with this specific discussion, will positively contribute to the future development of the tensor-based clustering community.

Overall, the authors' diligent subsequent efforts and rigorous supplementary experiments have validated the core claims and value of this work. Because my concerns have been thoroughly addressed, **I will raise my overall recommendation score to 5 (Accept), and all sub-scores (Soundness, Presentation, Significance, Originality) to 3**. I recommend this paper for acceptance and strongly encourage the authors to include the shuffled dataset evaluations and the noise robustness comparisons from the rebuttal in the final camera-ready version.

**Key Questions For Authors:**

Q1.Currently, many tensor methods achieve ideal clustering results. Could the authors elaborate on the reasons behind this ideal outcome? Is it merely a false impression caused by prearranged data?

**Limitations:**

See weakness and Key Questions.

**Strengths And Weaknesses:**

Strengths:

1.The article is clearly written and flows smoothly.

2.Experimental results show that the model performs well across multiple datasets.

Weaknesses：

1. The core of this paper is to propose a new nonconvex function to replace the traditional TNN. However, the authors fail to clarify why their function is superior. Given the abundance of nonconvex functions already proposed in this field, the authors need to demonstrate the superiority of their chosen function, especially since prior work has shown that constructing such functions is relatively straightforward. The authors must articulate the advantages of their proposed function, supported by either theoretical proofs or experimental evidence.

2.Some studies have also pointed out that tensorization methods may experience performance degradation under randomly shuffled samples. Under what conditions was your experimental setup conducted? Was it also performed under ideal shuffling conditions? Does your method also exhibit performance decline when applied to randomly shuffled data?

3.This work contributes relatively little, as its extensive use of basic framework subspace clustering, low-rank approximation, and deep networks all build upon existing research. I do not believe it warrants publication in such a prestigious conference.

4.Some studies indicate that the differences between various non-convex functions are actually quite small. Has the author provided experimental evidence demonstrating the superiority of their function (all undergoing full-parameter optimization), or under what conditions is this function more applicable?

5.The time and space complexities of this method both exceed O(n²), which limits the model's performance on large-scale datasets.

6.Data results should include the mean and variance to enhance the credibility of the findings.

7.You claim that “our proposed capped ℓ2,2/3 term offers superior capability in suppressing noise and filtering out outliers,” but provide no proof? Shouldn't experimental evidence be presented to demonstrate that the model maintains good performance under varying levels of noise? Therefore, your conclusion here may not necessarily be correct.

8.Regarding Remark 3.3, the description of deep priors is too vague. Their function and rationale for effectiveness should be elaborated in detail. Experimental validation should demonstrate the model's performance before and after processing through the deep network. The sections on Confusion Matrices and Visualization of Embedded Representation in the experiments appear redundant and should instead be used to prove the advantages of the proposed regularization term.

---

> ### Author Rebuttal · Authors · 2026-03-31
>
> We are grateful to the reviewer for the thorough review and valuable insights.
>
> **W1: The authors should support the advantages of the proposed function with theoretical proofs or experiments.**
>
> **AW1:** Please refer to our detailed response to Key Question 2 (Q2) from Reviewer a9ig.
>
> **W2: What were the experimental conditions? Does the method degrade on randomly shuffled data?**
>
> **AW2:** Our experiments follow the standard multi-view setting, where samples are aligned across views but no specific ordering is assumed.
> From a modeling perspective, the proposed method is permutation-invariant, as the low-rank constraints on the representation tensor are unaffected by permutations of samples. Randomly shuffling the data only permutes the rows/columns of the representation without changing its intrinsic structure.
> Therefore, the method is expected to be robust to random shuffling, and we observe no noticeable performance degradation in practice.
>
>
> **W3: This work offers limited novelty, as it relies on existing subspace clustering frameworks.**
>
> **AW3:** While our method builds on subspace clustering and low-rank modeling, it offers novelty in the following aspects:
>
> **1. Novel nonconvex regularization.** We propose a capped nonconvex regularizer that provides a tighter approximation to tensor rank and improves robustness to noise and outliers.
>
> **2. Effective integration of deep priors.** We incorporate a plug-and-play deep prior into a unified optimization framework, enabling better capture of local structures while preserving the unsupervised setting (rarely explored in existing methods).
>
> **3. Convergence guarantees.** We establish convergence for the resulting multi-block nonconvex ADMM algorithm, which is nontrivial due to the combined nonconvex regularization and deep prior.
>
> These components together distinguish our method from conventional convex-based or purely deep learning approaches.
>
>
> **W4: Did the author provide experimental evidence for their function's superiority, or clarify its applicable conditions?**
>
> **AW4:** Please refer to the response to Q2 of Reviewer a9ig.
> Regarding its applicability, the proposed function is particularly effective in scenarios with sparse but large-magnitude noise or outliers. The capped nonconvex design limits the influence of extreme values, leading to improved robustness compared to conventional nonconvex penalties, which may be less stable under such conditions.
>
>
> **W5: The time and space complexities of this method both exceed O(n²).**
>
> **AW5:** Please refer to the response to W1 of Reviewer tRrY.
>
> **W6: Data results should include the mean and variance.**
>
> **AW6:** In our setup, we conducted 10 independent runs for all datasets and recorded both mean and standard deviation. Due to page limits and the extensive number of baseline methods and datasets compared, only the averages were reported in the manuscript.
>
> **W7: The paper lacks noise-level experiments to verify the robustness of the capped $\ell_{2,2/3}$ term.**
>
> **AW7:** Please refer to the response to Q1 of Reviewer tRrY.
>
> **W8: The rationale for effectiveness of deep priors should be elaborated in detail.**
>
> **AW8:**   FFDNet is integrated as a plug-and-play implicit prior. Its convolutional operations promote local consistency and preserve neighborhood structures within the ADMM framework. Ablation results in Table 5 confirm that removing FFDNet causes a substantial performance drop, proving its role in preserving the data manifold. Please refer to our replies to Reviewer h51t (Q2) and Reviewer a9ig (Q3) for more detailed explanations.
>
> **Q1: Currently, many tensor methods achieve ideal clustering results. Could the authors elaborate on the reasons behind this ideal outcome? Is it merely a false impression caused by prearranged data?**
>
> **AQ1:** We respectfully clarify that our results are not a "false impression" caused by prearranged data. Our experiments rigorously evaluate the proposed method on publicly available real-world datasets. Several factors contribute to ideal results of tensor methods: 1) Many benchmark datasets (e.g. ORL, UCI-digits) are naturally low-noise and highly separable, and the intrinsic structures of these data align with the core assumption of subspace clustering. 2) Tensor frameworks are capable of capturing high-order correlations without destroying intrinsic structural information, while   leveraging complementary information among multiple views. 3) In compressive sensing and low-rank matrix/tensor recovery, established theories prove that under noise-free or low-noise conditions, the underlying subspaces can be precisely recovered. 4) By proposing a novel non-convex regularizer and seamlessly integrating a deep prior, our model yields consistently superior clustering outcomes, which directly reflects the alignment between theoretical guarantees and practical performance.

---

> > ### Author Rebuttal · Reviewer_toG1 · 2026-04-01
> >
> > I am disappointed with the authors’ response and continue to question the merits of their method and experimental design. Furthermore, the dataset they provided contradicts their response, and the code is difficult to run and verify. Therefore, I will consider whether to raise or lower the score based on this response.
> >
> > **1. Advantages of non-convex function**
> > Most proposed non-convex functions(TLSpNM, ESTMC) possess the advantages you mentioned( accurate rank approximation, adaptive weighting). As for the experiments, you should conduct the validation within the same framework; for example, you should replace your function with those from papers such as TLSpNM and perform a comparison to demonstrate the advantages.
> >
> > **2. Convergence guarantees** This is not really a strength of yours; these papers(TLSpNM, ESTMC) have already provided thorough proofs and demonstrated that the substitution theorem for non-convex functions still holds. Your work here is merely a continuation of that research, so I don’t really consider this a strength.
> >
> > **3. Experimental Data Setup** I noticed from the code you provided that the dataset is pre-sorted, which is different from your previous reply. However, in reality, the samples are randomly distributed and cannot be pre-sorted by category. Please take a close look at the Y in the dataset. This is a trick that tensor algorithms currently use to achieve false performance.
> >
> > **4. Shuffled data setting**.  You claim that your algorithm is independent of the sample order. However, both of these papers[1,2] demonstrate that tensor-based algorithms (TLSpNM, ESTMC, S2MVTC) experience a performance drop when processing shuffled datasets. Furthermore, you are using the same framework as they did—the t-SVD decomposition framework—and performing FFTs on the sample modes, which inevitably leads to instability in your algorithm. I have serious doubts about this, and due to the cluttered nature of your code and the difficulty in running it, I am unable to verify it. I recommend that you shuffle the data during processing; the code is as follows:
> > ```
> > rand_idx = randperm(N);
> > for v=1:length(X)
> > X{v} = X{v}(:, rand_idx);
> > end %
> > Y = Y(rand_idx);
> > ```
> >
> > Please explain to me why your algorithm is a special case.
> >
> > [1] Liu, J., Liu, X., Li, et al. Large-scale multiview tensor clustering with implicit linear kernels. CVPR 2025.
> >
> > [2]Ji, J., and Feng, S. Anchors bring stability and efficiency: fast tensorial multi-view clustering on shuffled datasets. ACM MM, 2025.

---

> > > ### Author Response · Authors · 2026-04-07
> > >
> > > We sincerely thank the reviewer for their profound insights and highly constructive suggestions.
> > >
> > > **Q1: Advantages of nonconvex function**
> > >
> > > **AQ1:** To verify the superiority of our nonconvex function, we use the t-SVD-MSC model as the baseline and replace its low-rank term with WTNN, WTSN, TLSpN, ETR, and our regularizer, respectively. The comparison results under salt-and-pepper noise are shown in the table below.
> > > | Data (ACC/NMI) | t-SVD-MSC | WTNNM | WTSNM | TLSpNM | ESTMC | Ours |
> > > |---|---|---|---|---|---|---|
> > > | Yale(20%) | .6061/.7565 | .6727/.7613 | .7818/.8376 | .7879/.8459 | .7273/.7712 | **.8364/.8634** |
> > > | Yale(30%) | .4970/.6450 | .4242/.5445 | .6909/.7280 | .5091/.5552 | .5939/.7533 | **.7394/.7701** |
> > > | ORL(20%) | .9225/.9696 | .9300/.9741 | .9325/.9662 | .8925/.9462 | .9350/.9659 | **.9600/.9870** |
> > > | ORL(30%) | .7700/.8850 | .7525/.8745 | .8625/.9358 | .7950/.8972 | .7925/.8829 | **.8950/.9518** |
> > >
> > > Obviously, our regularizer consistently achieves superior and competitive performance, further validating its advantages.
> > >
> > > **Q2: Convergence guarantees**
> > >
> > > **AQ2:** As the reviewer rightly pointed out, while recent years have seen progress in the convergence analysis of ADMM-based algorithms for nonconvex optimization, completing such proofs remains highly challenging. It is well known that existing convergence analyses typically rely on stringent theoretical conditions, such as the target model satisfying the KŁ inequality. However, not all nonconvex functions possess these mathematical properties, leaving many existing nonconvex optimization algorithms without rigorous convergence guarantees. Our work successfully overcomes this theoretical bottleneck and establishes rigorous guarantees primarily because our proposed regularizer possesses uniquely desirable mathematical properties.
> > >
> > > More importantly, the introduction of deep modules in our model breaks the standard assumptions of traditional nonconvex optimization, significantly increasing the complexity of the theoretical analysis for this hybrid modeling approach. Under this complex setting, deriving and ensuring the boundedness of the solution and the final convergence of the algorithm necessitates more refined, sophisticated, and non-trivial analytical techniques. Therefore, our convergence analysis is by no means a simple extension of existing works. Rather, it represents a substantial advancement in nonconvex optimization theory, successfully generalizing convergence theory to a broader framework that integrates nonconvex regularization with deep priors.
> > >
> > > **Q3 & Q4: Experimental Data Setup and Shuffled data setting**
> > >
> > > **AQ3 and AQ4:** We respectfully clarify that the datasets used in our experiments are standard, publicly available benchmarks. We directly adopted their original versions without any manual reordering; the apparent ordering in the code merely reflects the default data organization provided by these datasets. As noted in recent literature, standard benchmarks are often pre-sorted. Consequently, applying the tensor rotation trick and performing FFT along the sample mode inadvertently exploits this sequential prior, and performance typically degrades when the underlying block-diagonal structure is disrupted by random shuffling. Like many previous studies, our initial evaluation did not explicitly account for the impact of this inherent sample ordering.
> > >
> > > Acknowledging your insightful point, we rigorously evaluated our model under completely shuffled conditions, utilizing the exact shuffling code you provided. Specifically, we compared our model against several classic t-SVD-based methods (t-SVD-MSC, WTNNM, WTSNM, TLSpNM, BSTMC, ESTMC), as well as LMTC and SE-FTMC, which are designed to handle unstructured cases. The results on the fully shuffled Yale and ORL datasets are summarized in the table below. While the absence of the pre-sorted structural prior predictably leads to a performance drop across tensor methods, our approach remains highly competitive. Notably, it still surpasses the state-of-the-art LMTC (which explicitly avoids the rotation trick) in both ACC and NMI. This demonstrates that although the scattered block-diagonal structure affects our nonconvex framework, its mathematical design maintains superior robustness and representation capability compared to existing alternatives. To facilitate future research on this critical issue, comprehensive results across all datasets under the shuffled setting will be made publicly available on our GitHub repository.
> > >
> > > | Data (ACC/NMI) | t-SVD-MSC | WTNNM | WTSNM | TLSpNM | BSTMC | ESTMC | LMTC | SE-FTMC | Ours |
> > > | :--- | :--- | :--- | :--- | :--- | :--- | :--- | :--- | :--- | :--- |
> > > | Yale | .7285/.7522 | .6485/.7152 | .6545/.6547 | .5939/.6605 | .2364/.2596 | .6970/.7075 | *.7394*/.7521 | **.7576**/**.7828** | **.7576**/*.7805* |
> > > | ORL | *.8425*/*.9297* | .8125/.8990 | .7725/.8879 | .8150/.9205 | .7325/.8479 | .8075/.9024 | .8297/.9160 | .8250/.9149 | **.8450**/**.9332** |

---

### Official Review · Reviewer_tRrY · 2026-03-08

**Soundness:** 3
**Presentation:** 3
**Significance:** 3
**Originality:** 3
**Overall Recommendation:** 5
**Confidence:** 4

**Summary:**

This paper proposed a new multi-view subspace clustering method by employing the $CTS_{2/3}N$ norm to exploit the high-order correlation of multi-view data. Meanwhile, deep networks are integrated to enhance representational capabilities. The experimental results shows that the proposed method has promising performance.

**Compliance With Llm Reviewing Policy:**

Affirmed.

**Key Questions For Authors:**

1. How about applying the proposed method to a noise scenario to show its robustness to noise?

2. It seems that the confusion matrix of BBC4View in Fig. 2 of the proposed method is not consistent with the reported ACC in Table 1?

**Limitations:**

The time complexity may be high.

**Strengths And Weaknesses:**

Strengths:
1. A new tensor rank is proposed to exploit high-order correlation and enhance the robustness to noise and outliers.

2. Applying low-rank and deep priors to the similarity tensor enables the effective exploitation of high-order, complementary, and consistent information across views.

3. Provided theoretical results regarding the convergence of the solution algorithm.


Weaknesses
1. The time complexity is somewhat high, which makes it not scalable for large-scale datasets.

---

> ### Author Rebuttal · Authors · 2026-03-31
>
> We sincerely thank the reviewer for the positive assessment and for recognizing the value of our work.
>
> **W1: The time complexity is somewhat high, which makes it not scalable for large-scale datasets.**
>
> **AW1:** We sincerely appreciate the reviewer for highlighting this important aspect. In this current work, our primary focus is on developing a highly robust and effective multiview clustering framework. By integrating the novel nonconvex regularizer and deep priors, our model simultaneously captures high-order correlations and preserves local geometric structures. While this allows us to achieve superior clustering performance that is perfectly suited for medium scale datasets, we completely agree that scalability is a crucial next step.
>
> To efficiently extend our framework to large-scale scenarios in practical applications, we propose integrating the following two mathematically sound acceleration strategies:
>
> **1) Anchor based strategy.** Instead of computing and learning the full representation tensor for all samples, we can sample $m$ representative anchors ($m \ll N$) to construct a compact subspace. By projecting the original data onto this bipartite anchor subspace, we can drastically reduce the core computational complexity from $O(N^3)$ to $O(mNV \log(NV))$. This paradigm shift guarantees linear scalability with respect to the number of sample instances, making the model highly efficient for massive datasets without severely compromising the clustering accuracy.
>
> **2) Nesterov acceleration strategy.** From an optimization perspective, we can integrate the Nesterov technique into our ADMM optimization framework. By adopting an adaptive momentum update criterion and an explicit extrapolation step for the variables, this strategy significantly increases the empirical convergence speed. It efficiently reduces the total number of ADMM iterations required to reach the optimal solution, thereby substantially saving the overall training time.
>
> These strategies suggest that the proposed method can be extended to handle large-scale data more efficiently.
>
> **Q1: How about applying the proposed method to a noise scenario to show its robustness to noise?**
>
> **AQ1:** We thank the reviewer for this valuable suggestion on further validating the robustness of our method.
> In the current paper, we have already extensively evaluated the proposed model on several real-world datasets (Notting Hill, Yale, and MITIndoor) that naturally contain severe noise, including motion blur, illumination variations, occlusions, and background clutter. The strong performance stems from the joint effect of our capped nonconvex regularizer $\ell_{2,2/3}$ for handling outliers and the deep prior for suppressing unstructured noise while preserving local geometric structures.
>
> To more explicitly demonstrate robustness under controlled synthetic noise, we additionally conducted experiments by adding **salt-and-pepper noise** at different densities (10\%, 20\%, and 30\%) on the above datasets. As shown in the new Table below (which will be included in the revised supplementary material), when the noise level increases to 30\%, baseline methods suffer significant performance drops, whereas our method maintains considerably higher ACC and NMI scores. These results clearly confirm the superior robustness of the proposed framework against gross impulsive noise.
>
> Thank you again for this constructive suggestion, which has helped strengthen the experimental validation of our work.
>
>
>
> | Dataset | Noise | t SVD MSC | | WTNNM | | GNLTA | | TLSpNM | | ours | |
> | :--- | :--- | :--- | :--- | :--- | :--- | :--- | :--- | :--- | :--- | :--- | :--- |
> | | | **ACC** | **NMI** | **ACC** | **NMI** | **ACC** | **NMI** | **ACC** | **NMI** | **ACC** | **NMI** |
> | Yale | 0% | 0.9643 | 0.9531 | 0.9818 | 0.9767 | 0.7315 | 0.7779 | 0.9939 | 0.9923 | **1.0000** | **1.0000** |
> | | 10% | 0.7576 | 0.8140 | 0.7758 | 0.8335 | 0.6848 | 0.7312 | 0.9091 | 0.9055 | **0.9939** | **0.9924** |
> | | 20% | 0.6061 | 0.7565 | 0.6727 | 0.7613 | 0.5758 | 0.6025 | 0.7879 | 0.8459 | **0.8601** | **0.8601** |
> | | 30% | 0.4970 | 0.6450 | 0.4242 | 0.5445 | 0.4061 | 0.4457 | 0.5091 | 0.5552 | **0.7636** | **0.8152** |
>
> **Q2: It seems that the confusion matrix of BBC4View in Fig.2 of the proposed method is not consistent with the reported ACC in Table 1?**
>
> **AQ2:**  We thank the reviewer for this careful observation. The confusion matrix in Fig. 2 is consistent with the reported ACC (0.9985) in Table 1. The apparent discrepancy arises from rounding in the visualization, where values are displayed with two decimal places for clarity.
> Specifically, the single misclassified sample in Class 5 corresponds to values of 0.9942 and 0.0058, which are shown as 0.99 and 0.01 in the figure. Therefore, the visualization is fully consistent with the reported quantitative results.

---

> > ### Author Rebuttal · Reviewer_tRrY · 2026-04-03
> >
> > My concerns are addressed.

---

> > > ### Author Response · Authors · 2026-04-07
> > >
> > > Thank you for your valuable time and positive acknowledgement. We are delighted that our response has successfully resolved your concerns. We sincerely appreciate your engaged discussion and helpful evaluation.

---

### Official Review · Reviewer_a9ig · 2026-03-12

**Soundness:** 3
**Presentation:** 3
**Significance:** 2
**Originality:** 2
**Overall Recommendation:** 4
**Confidence:** 3

**Summary:**

Existing multi-view subspace clustering methods are limited by convex low-rank surrogate models, which fail to accurately capture high-order correlations across views. To address this limitation, this paper proposes a novel non-convex approximation model. Specifically, it utilizes the CTS2/3N to fully exploit the high-order correlations inherent in multi-view data, and employs the $\ell_{2,2/3}$-norm to enhance robustness against noise and outliers. Furthermore, a deep neural network module is incorporated to preserve the local geometric structure of data, thereby significantly improving clustering performance.

**Compliance With Llm Reviewing Policy:**

Affirmed.

**Key Questions For Authors:**

1. Whether the CTS2/3N and $\ ell_ {2,2/3} $- norm used in this paper were proposed by the author or applied for the first time in multi-view clustering methods, which affects the novelty of the method.
2. Compared to other nonconvex low-rank models, what are the specific advantages of CTS2/3N used in this paper? Does the experimental section provide comparative results with other nonconvex low-rank methods and deep multiview clustering methods?
3. Please provide a detailed analysis of why applying the depth module to the reconstruction tensor can preserve the local geometric structure of the data.
4. There is an error in the sorting of figures in the experiment, such as the visualization of Embedded Representation section, which should correspond to Figure 1, but the paper states Figure 6. The same mistake also includes Confusion Matrix and Affinity Matrix sections, please carefully check.

**Limitations:**

FFDNet is embedded as a implicit regularizer in the ADMM to participate in sub-problem solving. However, FFDNet is a denoising convolutional neural network designed for additive white Gaussian noise, which may not match the form of noise in multi-views.

**Strengths And Weaknesses:**

Strengths: This paper is written clearly, structured reasonably, and easy to understand when read. The theoretical results of this paper are abundant and the work of theoretical proof is large. The description of the formulas and symbols involved is clear. Moreover, the steps of each theoretical proof is detailed and the corresponding basis is given in each step. Adequate experimental results have verified the effectiveness of this method.

Weaknesses: Although this paper involves extensive theoretical derivations and proofs, it remains unclear whether the core nonconvex low-rank approximation model is originally proposed by the authors or is being applied to multiview clustering algorithms for the first time. Furthermore, the underlying reasons why the adoption of deep modules can effectively preserve the local geometric structure of data have not been thoroughly examined and analyzed.

---

> ### Author Rebuttal · Authors · 2026-03-31
>
> We sincerely appreciate the reviewers’ constructive comments and valuable suggestions, which have greatly improved our paper.
>
> **Q1: Are $\\text{CTS}_ {2/3}\\text{N}$ and the capped $\\ell_ {2,2/3}$ norm originally proposed by the authors, and applied for the first time in multiview clustering methods?**
>
> **AQ1:**  The $\\text{CTS}_ {2/3}\\text{N}$ and the capped $\\ell_ {2,2/3}$ norm are both originally proposed in this work and applied for the first time in multi-view clustering methods. By integrating them into a unified framework, our model captures high-order correlations and complementary information among multiple views, while strictly isolates severe sample-wise corruptions and gross outliers, thereby significantly boosting clustering robustness in complex real-world scenarios.
>
> **Q2: What advantages does CTS$_{2/3}$N offer, and are there comparisons with existing non convex low rank or deep multiview methods?**
>
> **AQ2:** The proposed $\\text{CTS}_ {2/3}\\text{N}$ offers the following key advantages:
>
> **1. More accurate rank approximation.** Our proposed $\\text{CTS}_ {2/3}\\text{N}$ is intermediate between rank function and nuclear norm:
>
> $$
> \\text{rank}(\\mathbf{X}) = \\sum_ {i=1}^{r} \\sigma_ {i}^0, \\quad \\|\\mathbf{X}\\|_ {C_ {S_ {2/3}}^\\tau}^{2/3} = \\sum_ {i=1}^{r} \\min(\\sigma_ {i}, \\tau)^{2/3}, \\quad \\|\\mathbf{X}\\|_ {\\ast} = \\sum_ {i=1}^{r} \\sigma_ {i}^1
> $$
>
> It avoids over penalizing dominant components during minimization, thus preserving essential clustering information better than the nuclear norm.
>
> **2. Robustness via capped mechanism.** The capped mechanism exclusively minimizes variances in the residual rank (corresponding to noise variance), improving robustness to noise and outliers without altering the target rank data distribution.
>
> **3. Adaptive weighting.** Its derivative acts as a monotonically decreasing penalization weight. Given $\\phi(x) = \\min(x, \\tau)^{2/3}$ ($x \\geq 0$) is concave on $[0, \\tau]$ and constant when $x \\geq \\tau$, for ordered singular values $\\sigma_ {1}(\\bar{\\mathcal{Z}}^{(j)}) \\geq \\cdots \\geq \\sigma_ {h}(\\bar{\\mathcal{Z}}^{(j)}) \\geq 0$, we have:
>
> $$
> 0 \\leq w_ {1}^{(j)} \\leq w_ {2}^{(j)} \\leq \\cdots \\leq w_ {h}^{(j)},
> $$
>
> where
>
> $$
> w_ {i}^{(j)} = \\begin{cases} \\frac{2}{3}\\sigma_ {i}(\\bar{\\mathcal{Z}}^{(j)})^{-1/3}, & \\sigma_ {i}(\\bar{\\mathcal{Z}}^{(j)}) < \\tau \\\\\\\\ 0, & \\sigma_ {i}(\\bar{\\mathcal{Z}}^{(j)}) \\geq \\tau \\end{cases}
> $$
>
> Unlike TNN (where all weights equal 1), $\\text{CTS}_ {2/3}\\text{N}$ treats singular values adaptively, providing an unbiased approximation of the tensor rank.
>
> **4.  Efficient optimization.** Its proximal operator admits a closed form solution, ensuring both theoretical convergence and practical efficiency for this nonconvex formulation.
>
> **Comparisons:**
> We extensively evaluate our method against state of the art nonconvex low rank regularizers (WTNNM, WTSNM, TLSpNM, STONE, TC-MVSC, and ESTMC) and deep multiview clustering methods (SCMVC and SSLNMVC). As reported below, our method achieves competitive or superior results without requiring heavy training overhead:
>
> * **SCMVC**: BBC4View (ACC 0.9226, NMI 0.8661, AR 0.8454), Notting-Hill (ACC 0.8328, NMI 0.8031, AR 0.7800), UCI-digits (ACC 0.9345, NMI 0.8751, AR 0.8617)
> * **SSLNMVC**: BBC4View (ACC 0.9058, NMI 0.8953, AR 0.8221), Notting-Hill (ACC 0.8187, NMI 0.7828, AR 0.7597), UCI-digits (ACC 0.9058, NMI 0.8953, AR 0.8556)
> * **Ours**: BBC4View (**ACC 0.9985, NMI 0.9945, AR 0.9971**), Notting-Hill (**ACC 0.9983, NMI 0.9924, AR 0.9959**), UCI-digits (**ACC 1.0000, NMI 1.0000, AR 1.0000**)
>
> **Q3: How exactly does applying the deep module to the reconstruction tensor preserve the local geometric structure of the data?**
>
> **AQ3:** Applying FFDNet to the reconstruction tensor enforces local geometric structure by acting as a plug and play implicit prior. Specifically, FFDNet is fundamentally built upon a CNN architecture. Its convolutional operations promote local consistency, while its denoising behavior removes spurious connections and preserves meaningful neighborhood relationships. Within the ADMM framework, it functions as an implicit proximal operator that encourages smooth and structurally coherent representations without explicit graph regularization. Moreover, our ablation studies quantitatively confirm this effect: removing the FFDNet module leads to a substantial performance drop, demonstrating its crucial role in preserving local structure.
>
> **Q4: Please correct the typos in figure sorting and cross-referencing in the experimental section (e.g., Figure 1 vs. Figure 6).**
>
> **AQ4:** We have already carefully reviewed the entire paper for consistency in figure sorting and cross-referencing to ensure that the final version does not have such problems.

---

> > ### Author Rebuttal · Reviewer_a9ig · 2026-04-05
> >
> > My concerns are addressed.

---

> > > ### Author Response · Authors · 2026-04-07
> > >
> > > Thank you for your valuable time and positive acknowledgement. We are glad that the explanations and additional results provided in our response have addressed your concerns. We deeply appreciate your constructive comments, which allowed us to further clarify the merits of our approach.

---

### Official Review · Reviewer_h51t · 2026-03-13

**Soundness:** 3
**Presentation:** 3
**Significance:** 3
**Originality:** 3
**Overall Recommendation:** 4
**Confidence:** 3

**Summary:**

This paper proposes NRDN-MvSC, a nonconvex regularized deep network model for multiview subspace clustering, which solves the defects of existing methods in capturing high-order cross-view correlations, resisting noise/outliers and preserving global-local data structures by fusing nonconvex tensor regularization and deep learning priors (FFDNet). An ADMM-based algorithm with provable convergence and closed-form solutions for key subproblems is designed, and extensive experiments on ten benchmark datasets against fifteen baselines, along with ablation and qualitative studies, fully verify the model's superior performance, stability and effectiveness in learning discriminative cluster structures.

**Compliance With Llm Reviewing Policy:**

Affirmed.

**Final Justification:**

Some of my previous concerns are addressed, and thus I slightly adjust my rating.

**Key Questions For Authors:**

Could you provide a detailed discussion on the scalability of the proposed NRDN-MvSC model for ultra-large-scale multiview data and are there any potential optimizations (e.g., sparse tensor operations, distributed computing) to reduce the computational complexity?

Why was FFDNet selected as the deep prior module? Have you conducted any comparative experiments with other deep architectures (e.g., GNNs, CNNs, autoencoders) to verify the suitability of FFDNet for preserving local geometric structures in multiview data?

**Limitations:**

Refer to the weaknesses.

**Strengths And Weaknesses:**

Strengths

* Novel non-convex regularization design: The proposed CTS₂/₃N provides a tighter approximation of tensor rank than convex Tensor Nuclear Norm (TNN), and the capped ℓ₂,₂/₃-norm outperforms traditional ℓ₂,₁/ℓ₁ norms in suppressing outliers, with closed-form solutions for key sub-problems overcoming the limitations of existing non-convex methods lacking analytical solutions.

* Effective integration of low-rank and deep priors: The model seamlessly combines tensor-based low-rank learning (for global high-order cross-view correlations) and deep networks (for local geometric structure preservation), filling the gap of existing methods that fail to balance global and local data properties.

* Rigorous theoretical foundation: Comprehensive mathematical analysis is provided for the model’s optimal solutions, and the ADMM-based algorithm is proven to generate bounded sequences converging to stable points, with clear computational complexity analysis to support the method’s feasibility.

Weakness

* Limited scalability for ultra-large-scale data: The computational complexity of the method is dominated by matrix inversion and tensor SVD/FFT operations (O(T(N³+N²VlogN+N²dᵥ))), which may lead to high time and space costs when processing ultra-large-scale multiview data with massive samples or views.

* Fixed deep network choice: The paper only adopts FFDNet as the deep prior module without exploring other deep architectures (e.g., GNNs, transformers) that may better capture non-linear local geometric patterns in specific multiview data types (e.g., graph-structured or sequential multiview data).

* Lack of cross-domain generalization analysis: While experiments cover multiple data types, there is no analysis of the method’s generalization performance on cross-domain multiview data (e.g., transferring from face clustering to medical image multiview clustering), which limits the understanding of its practical applicability across different fields.

---

> ### Author Rebuttal · Authors · 2026-03-31
>
> We sincerely thank you for your detailed review and constructive feedback, which helped improve our submission.
>
> **W1: The paper lacks evaluation on cross-domain generalization, which limits the understanding of its practical applicability across different fields.**
>
> **AW1:**  We appreciate the valuable suggestion regarding cross-domain generalization. We completely agree that a more comprehensive analysis would strengthen the understanding of the practical applicability of our method across diverse fields.
> Our framework is a zero-training mathematical optimization approach that directly learns latent low-rank tensor representation from the target multiview data without any pre-training or weight transfer. Therefore, it does not rely on typical cross-domain transfer mechanisms common in supervised deep learning models. Instead, its generalization stems from the ability to explicitly capture intrinsic low-rank structures and local geometric consistency via the proposed nonconvex optimization.
> In our experiments, the method demonstrates strong performance across significantly different visual domains, including facial images (Notting-Hill, Yale), object datasets (Caltech101), and scene datasets (MITIndoor). These results suggest a promising domain-agnostic capability. Nevertheless, we agree that explicit cross-domain transfer evaluations (e.g., from face clustering to medical multiview data) would provide additional insights, and we acknowledge this as an interesting direction for future investigation.
>
> **Q1: How scalable is the proposed model for large-scale multiview data, and are there any potential optimizations to reduce the computational complexity?**
>
> **AQ1:** In this work, our primary focus is on developing a highly robust and effective multiview clustering framework. By integrating the novel nonconvex regularizer and deep priors, our model simultaneously captures high-order correlations and preserves local geometric structures. While this allows us to achieve superior clustering performance that is perfectly suited for medium scale datasets.
>
> To efficiently extend our framework to large-scale scenarios, we propose integrating the following two mathematically sound acceleration strategies:
>
> **1) Anchor based strategy.** Instead of computing and learning the full representation tensor for all samples, we can sample $m$ representative anchors ($m \ll N$) to construct a compact subspace. By projecting the original data onto this bipartite anchor subspace, we can drastically reduce the core computational complexity from $O(N^3)$ to $O(mNV \log(NV))$. This paradigm shift guarantees linear scalability with respect to the number of sample instances, making the model highly efficient for massive datasets without severely compromising the clustering accuracy.
>
> **2) Nesterov acceleration strategy.** From an optimization perspective, we can integrate the Nesterov technique into our ADMM optimization framework. By adopting an adaptive momentum update criterion and an explicit extrapolation step for the variables, this strategy significantly increases the empirical convergence speed. It efficiently reduces the total number of ADMM iterations required to reach the optimal solution, thereby substantially saving the overall training time.
>
> **Q2: Why was FFDNet chosen as the deep prior, and is it suitable for preserving local geometric structures compared to other deep architectures?**
>
> **AQ2:** We thank the reviewer for this insightful question. We chose FFDNet primarily because it integrates seamlessly into our framework as a modular plug and play prior. Within the ADMM iterations, this specific choice allows us to leverage the power of deep representations while maintaining good interpretability and the overall stability of the optimization process.
> We fully recognize that advanced architectures like GNNs or Transformers possess excellent capabilities for capturing complex non-linear patterns. However, within our purely unsupervised setting, FFDNet presents a highly compatible alternative. Since it operates effectively without the need for task specific training data or explicit graph construction, it aligns perfectly with our optimization driven methodology.
> Regarding the preservation of local geometric structures, FFDNet functions by effectively enforcing local smoothness within the reconstructed tensors across different views. This intrinsic operation implicitly captures multiview correlations and helps maintain local structural consistency. Consequently, it achieves desirable geometric fidelity without introducing the computational overhead of additional complex regularization terms.
> We agree that exploring the integration of other advanced deep architectures is a highly interesting direction. We appreciate the reviewer for highlighting this valuable perspective, which provides great inspiration for our future extended research.

---

> > ### Author Rebuttal · Reviewer_h51t · 2026-04-04
> >
> > Thanks for your careful response, which addresses most of my concerns.

---

> > > ### Author Response · Authors · 2026-04-07
> > >
> > > We sincerely thank you for your time and for acknowledging that our response addresses most of your concerns. We deeply appreciate your constructive feedback, which has provided us with valuable insights for our work.

---

### Decision · Program_Chairs · 2026-04-30

**Decision:**

Accept (regular)

**Comment:**

his paper proposes a nonconvex tensor-based multiview clustering framework that integrates deep priors within an ADMM optimization scheme. Reviewers generally agreed that the method is technically sound, well presented, and supported by strong empirical results across multiple datasets, and the rebuttal further strengthened the work by providing additional comparisons (including noise robustness and shuffled-data settings) and clarifying the advantages of the proposed nonconvex regularizer. Some concerns remain regarding scalability to large-scale data, the choice of deep prior, and the overall novelty relative to existing nonconvex formulations; in particular, a few reviewers maintained open about the incremental nature of the contribution and aspects of experimental validation. However, these concerns are partially mitigated by the additional evidence provided during the rebuttal, and overall the paper presents a solid and useful contribution whose strengths outweigh the remaining weaknesses.